# Hierarchically branched diffusion models leverage dataset structure for class-conditional generation

**Alex M. Tseng**                                                        *tseng.alex@gene.com*
**Max Shen**                                                             *shen.max@gene.com*
**Tommaso Biancalani**                                                   *biancalt@gene.com*
**Gabriele Scalia**                                                      *scalia.gabriele@gene.com*
*Biology Research / AI Development*
*Genentech*

**Reviewed on OpenReview:** *https://openreview.net/forum?id=sGTfxqRbei*

## Abstract

Diffusion models have attained state-of-the-art performance in generating realistic objects, including when conditioning generation on class labels. Current class-conditional diffusion models, however, implicitly model the diffusion process on all classes in a flat fashion, ignoring any known relationships between classes. Class-labeled datasets, including those common in scientific domains, are rife with internal structure. To take advantage of this structure, we propose hierarchically branched diffusion models as a novel framework for class-conditional generation. Branched diffusion models explicitly leverage the inherent relationships between distinct classes in the dataset to learn the underlying diffusion process in a hierarchical manner. We highlight several advantages of branched diffusion models over the current state-of-the-art methods for class-conditional diffusion. Firstly, they can be easily extended to novel classes in a continual-learning setting at scale. Secondly, they enable more sophisticated forms of conditional generation, such as analogy-based conditional generation (i.e. transmutation). Finally, they offer a novel interpretability into the class-conditional generation process. We extensively evaluate branched diffusion models on several benchmark and large real-world scientific datasets, spanning different data modalities (images, tabular data, and graphs). We particularly highlight the advantages of branched diffusion models on a single-cell RNA-seq dataset, where our branched model leverages the intrinsic hierarchical structure between human cell types.

## 1 Introduction

Diffusion models have gained major popularity as a method for generating data from complex data distributions (Sohl-Dickstein et al., 2015; Ho et al., 2020; Song et al., 2021). Furthermore, they have also been successful in performing *conditional generation*, where we wish to sample some object $x$ conditioned on a label $y$. Recent works in conditional diffusion have arguably received the most popular attention (Song et al., 2021; Dhariwal & Nichol, 2021; Rombach et al., 2022; Ho et al., 2022), and have rapidly become a staple in generative AI.

Current diffusion models, however, are limited in their treatment of class-labeled datasets. Conventional diffusion models learn the diffusion process flatly for each class, disregarding any known relationships or structure between them. In reality, class-labeled datasets in many domains, such as those characterizing scientific applications, have an inherent structure between classes which can be thought of as hierarchical. For example, human cell types are organized hierarchically by nature: keratinocytes are very distinct from neurons, but the latter subdivide into excitatory and inhibitory neurons. Additionally, even when a dataset has no pre-defined hierarchical label set (e.g. an explicit ontology), hierarchical structure can often be found, because some subsets of classes are invariably more similar than others.

In order to leverage this intrinsic structure within these datasets, we propose restructuring diffusion models to be *hierarchically branched*, where the branching structure reflects the inherent relationships between classes in the dataset (the underlying diffusion process remains unchanged). By modeling the diffusion process in a hierarchical fashion, branched diffusion models enjoy several advantages which make them much more suitable in many applications, particularly in scientific settings. We apply branched diffusion to four different datasets, including two large real-world scientific datasets, spanning different data modalities (images, tabular data, and graphs) and showcase the following advantages over the current state-of-the-art method for conditional diffusion:

- Branched models offer a novel way to perform class-conditional generation via diffusion by organizing labels hierarchically.

- They can be easily extended to generate new, never-before-seen data classes in a continual-learning setting.

- They can be used to perform more sophisticated forms of conditional generation, such as analogy-based conditional generation (or "transmutation").

- Diffusing across their branched structure offers interpretability into the relationship between classes from a generative perspective, such as elucidating shared high-level features.

## 2 Related Work

Most diffusion models today are defined by a pair of forward and reverse stochastic differential equations (SDEs). The forward equation injects random noise over continuous time to transform the initial data distribution $p_0(x)$ into a tractable prior distribution $\pi(x)$ from $t = 0$ to the time horizon $t = T$:

$$dx = f(x,t)dt + g(t)d\omega, \tag{1}$$

where $\omega$ is a standard Wiener process (i.e. Brownian motion). $f(x,t)$ and $g(t)$ are the drift and diffusion coefficients, respectively. For the variance-preserving SDE, for example, $f(x,t) = -\frac{1}{2}\beta_t x$ and $g(t) = \sqrt{\beta_t}$ for some noise schedule $\beta_t$ (yielding a prior $\pi(x)$ which is an isotropic Gaussian).

In order to sample an object from $p_0(x)$, we first tractably sample from $\pi(x)$, and then follow the associated reverse SDE to recover a sample from $p_0(x)$:

$$d\mathbf{x} = \left[ f(x,t) - g(t)^2 s(x,t) \right] dt + g(t)d\omega', \tag{2}$$

where $\omega'$ is a standard Wiener process in the reverse direction. $s(x,t) = \nabla_x \log(p_t(x))$ is the Stein score of $x$ at diffusion time $t$. A neural network is trained to predict $s_\theta(x,t) \approx s(x,t)$, made possible by defining $f(x,t)$ and $g(t)$ such that the true Stein score is tractably defined in closed form (such as with the variance-preserving SDE).

In order to perform *class-conditional* generation, the diffusion model needs to learn the conditional distribution of data for each class of the dataset. The current state-of-the-art method for class-conditional diffusion was proposed in Ho et al. (2021), termed "classifier-free conditional generation". In this method, the reverse-diffusion neural network is given the class label $c$ as an auxiliary input, which *guides* the generation of objects to specific classes:

$$dx = \left[ f(x,t) - g(t)^2 s_\theta(x,t,c) \right] dt + g(t)d\omega', \tag{3}$$

where $s_\theta(x,t,c)$ is the trained neural network which approximates the Stein score at $x$ and time $t$ for the conditional distribution of class $c$.

This method of conditional generation has achieved state-of-the-art performance in sample quality (Rombach et al., 2022; Ho et al., 2022), and in contrast to the previous method of classifier guidance (Song et al., 2021), it can be applied to both continuous- and discrete-time diffusion models.

Note that the term "hierarchical diffusion" is somewhat overloaded, as there exist other works which use this term, but describe vastly different methods than the one proposed here. For example, Qiang et al. (2023) describes applying distinct diffusion processes iteratively to generate coarse-grained features before fine-grained features. Lu et al. (2023) proposes latent-space diffusion where latent embeddings of various granularities and sizes are passed to the diffusion denoising model. In contrast, the "hierarchy" in our work refers to a hierarchy of similarity between classes in the original dataset.

## 3 Hierarchically branched diffusion models

Suppose our dataset consists of a set of classes $C$. For example, let us consider MNIST handwritten digits, where $C = \{0, 1, ..., 9\}$. We wish to leverage the fact that some classes are inherently more similar than others (e.g. the 4s and 9s in MNIST are visually more similar to each other than they are to 0s). As noise is progressively added to data, there is some point in diffusion time at which any two samples from two different classes are so noisy that their original class effectively cannot be determined; we call this point in time a *branch point*. A branch point is a property of two classes (and the forward diffusion process), and—importantly—the more similar the two classes are, the earlier the branch point will be.

These branch points underpin the main distinction between a branched diffusion model and a traditional linear one. We define the branch point between two classes as the earliest diffusion time point when objects of the two classes are sufficiently similar in distribution, such that reverse diffusion after this point could be predicted by the same neural-network model (without specifying class identity). More formally, we defined a criterion for measuring when the distribution of two noisy classes is sufficiently similar (Equation 8), based off of the well-established metric of energy distance. Mathematical justification and more details can be found in Appendix A. Note that branch-point discovery is performed once for the dataset at the beginning, and this proposed procedure's runtime is orders-of-magnitude smaller than the time taken to train the model. Alternatively, since branch points reflect the inherent similarity between classes of the dataset, they may also be entirely defined by domain knowledge (e.g. an ontology describing known similarities between cell types, or chemical classes of drug-like molecules).

Together, the branch points between all classes in $C$ naturally encode a *hierarchy* of class similarities (Figure 1a). This hierarchy separates diffusion time from a single linear track into a branched structure, where each branch represents the diffusion of a subset of classes, and a subset of diffusion times. For $|C|$ classes, there are $2|C| - 1$ branches. Each branch $b_i = (s_i, t_i, C_i)$ is defined by a particular diffusion time interval $[s_i, t_i)$ (where $0 \le s_i < t_i < T$) and a subset of classes $C_i \subseteq C$ (where $C_i \ne \emptyset$). The branches are constrained such that every class and time $(c, t) \in C \times [0, T)$ can be assigned to exactly one branch $b_i$ such that $c \in C_i$ and $t \in [s_i, t_i)$. The branches form a rooted tree starting from $t = T$ to $t = 0$. Late branches (large $t$) are shared across many different classes, as these classes diffuse nearly identically at later times. Early branches (small $t$) are unique to smaller subsets of classes. The earliest branches are responsible for generating only a single class.

Additionally, as opposed to a conventional ("linear", or "non-hierarchical") diffusion model which learns to reverse diffuse all classes and times using a single-task neural network, a branched diffusion model is implemented as a *multi-task neural network*. Specifically, each output task predicts reverse diffusion for a single branch (e.g. in an SDE-based diffusion framework (Song et al., 2021), each prediction head learns the Stein score for a specific branch) (Figure 1b). The multi-task architecture allows the model to learn the reverse-diffusion process distinctly for each branch, while the shared parameters allow the network to learn shared representations across tasks without an explosion in model complexity:

$$dx = \left[ f(x, t) - g(t)^2 s_\theta(x, t)_{[b_{c,t}]} \right] dt + g(t) d\omega', \tag{4}$$

where $b_{c,t}$ is the branch index corresponding to the class $c$ of object $x$, and $t$ is the diffusion time. Note that *the forward- and reverse-diffusion processes are identical to traditional diffusion models* (Equations 1–2). However, in contrast to traditional linear models where the neural network $s_\theta(x, t, c)$ takes in the class as an input to learn class-conditional distributions (Equation 3), the neural network for a branched model *explicitly*

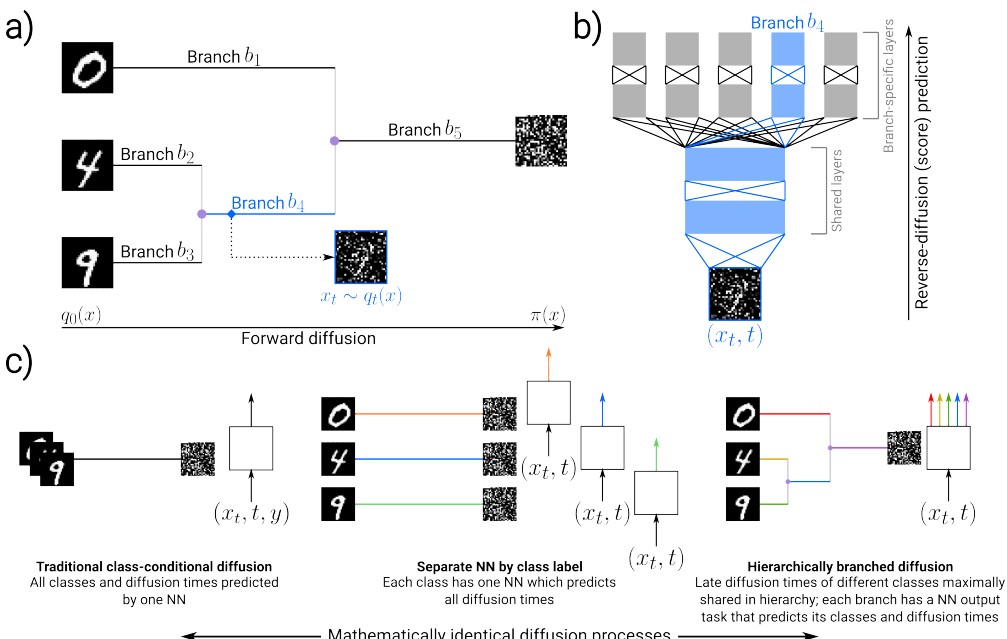

Figure 1: Schematic of a branched diffusion model. **a)** Some classes are more similar than others from a diffusion perspective: after adding sufficient noise, similar classes become indistinguishable from each other after *branch points*. Together, the branch points (purple dots) between all classes define a hierarchical structure. Although there is still only a single underlying diffusion process, the hierarchy separates diffusion time into branches, where each branch represents diffusion for a subset of classes and a subset of diffusion times. An example of one diffusion intermediate is highlighted in blue; this example is an MNIST digit that is a 4 or 9, and is at an intermediate diffusion time. **b)** A branched diffusion model is realized as a multi-task neural network (NN) that predicts reverse diffusion (one output task for each branch). The prediction path for the blue-highlighted MNIST digit in panel a) is also in blue. **c)** We show a progression of methods from traditional linear diffusion models to hierarchically branched models. *Left:* A traditional linear class-conditional diffusion model trains a single NN for all classes and diffusion times. Both class label and time are NN inputs. *Middle:* Consider a modification, where each class has its own NN, each of which is trained to reverse diffuse the full diffusion timeline for only its associated class. To generate an object of a specific class, we take reverse-diffusion predictions from the associated NN. *Right:* At late diffusion times, different classes diffuse very similarly. A branched diffusion model shares as much diffusion time as possible, thereby leading to a hierarchy of times, where each branch is learned by an output task of a multi-task NN. To generate a specific class, we take reverse-diffusion predictions from the appropriate subset of branches which cover the full diffusion timeline from $t = T$ to $t = 0$.

*learns the conditional distribution of each branch as a separate output task* (each branch has an associated output task with index $b_{c,t}$).

Training a branched diffusion model follows nearly the same procedure as with a standard linear model, except for each input, we only perform gradient descent on the associated branch, which corresponds to a model output task) (see Algorithm 1). To sample an object of class $c$, we perform reverse diffusion starting from time $T$ and follow the appropriate branch down (see Algorithm 2).

To better understand branched diffusion models and how they are different from traditional linear models, consider the following progression of methods (Figure 1c): **1)** A traditional class-conditional diffusion model, where all classes are treated in a flat fashion. A single neural network learns reverse diffusion for all classes and diffusion times; both the time and class label are inputs to the model (Equation 3). **2)** Instead of a single neural network, train a separate network for each data class (*the diffusion process is still the same for each class*). In contrast with method 1), we reverse diffuse a particular class by choosing the appropriate network. This would allow for benefits like class extension in continual learning (Section 5), but at the cost of inefficient parameterization and training time. Furthermore, fully separating diffusion timelines for each class into distinct neural networks ignores inherent class similarities, and as a result, this approach does not

---

**Algorithm 1** Training a branched diffusion model

---

**Input:** training set $\{(x^{(k)}, c^{(k)})\}$, branches $\{b_i\}$
**repeat**
    Sample $(x_0, c)$ from training data $\{(x^{(k)}, c^{(k)})\}$
    Sample $t \sim Unif(0, T)$
    Forward diffuse $x_t \sim q_t(x|x_0)$
    Find branch $b_i = (s_i, t_i, C_i)$ s.t. $s_i \leq t < t_i$, $c \in C_i$
    Gradient descent on $p_{(\theta_s, \theta_i)}(x_t, t)[i]$ (on output task $i$)
**until** convergence

---

---

**Algorithm 2** Sampling a branched diffusion model

---

**Input:** class $c$, trained $p_\theta$, branches $\{b_i\}$
Sample $\hat{x} \leftarrow x_T$ from $\pi(x)$
**for** $t = T$ to $0$ **do**
    Find branch $b_i = (s_i, t_i, C_i)$ s.t. $s_i \leq t < t_i$, $c \in C_i$
    $\hat{x} \leftarrow p_\theta(\hat{x}, t)[i]$ (take output task $i$)
**end for**
Return $\hat{x}$

---

allow for benefits like transmutation (Section 6) or interpretability of diffusion intermediates (Section 7). **3)** Owing to the inherent structure of the dataset, we can leverage the fact that noisy objects of different classes are indistinguishable after a certain diffusion time (i.e. branch points). Thus, we maximize sharing of diffusion time between classes via a hierarchy, and we train a multi-task neural network where each task predicts reverse diffusion for a single branch (Equation 4). We now generate a particular class by choosing the appropriate *set* of branches (output tasks). This allows us to retain benefits like class extension, and gain benefits like transmutation and interpretability, with a much more efficient parameterization and reduced training time compared to method 2).

Importantly, the underlying diffusion process and subsequent mathematical properties in a branched diffusion model are directly inherited from and *identical* to that of a conventional linear model. A branched model is characterized by the explicit definition of branch points which *separate* the responsibility of reverse diffusing different subsets of classes and times into separate branches, where each branch is predicted by a different head of a multi-task neural network.

## 4 Class-conditional diffusion via branched diffusion models

Branched diffusion models are a completely novel way to perform class-conditional diffusion. Instead of relying on external classifiers or labels as auxiliary neural-network inputs, a branched diffusion model generates data of a specific class simply by reverse diffusing down the appropriate branches.

We demonstrate branched diffusion models on several datasets of different data modalities: 1) MNIST handwritten-digit images (LeCun et al.); 2) a tabular dataset of several features for the 26 English letters in various fonts (Frey & Slate, 1991); 3) a real-world, large scientific dataset of single-cell RNA-seq, measuring the gene expression levels of many blood cell types in COVID-19 patients, influenza patients, and healthy donors (Lee et al., 2020); and 4) ZINC250K, a large dataset of 250K real drug-like molecules (Irwin et al., 2012). We trained continuous-time (i.e. SDE-based (Song et al., 2021)) branched diffusion models for all datasets. The branching structure was inferred by our branch-point discovery algorithm (Supplementary Tables S1–S8, Appendix A). We verified that our MNIST model generated high-quality digits (Supplementary Figure S1). For our tabular-letter dataset, we followed the procedure in Kotelnikov et al. (2022) to verify that the branched model generated realistic letters that are true to the training data (Supplementary Figure S2).

We compared the generative performance of our branched diffusion models to the current state-of-the-art methods for conditional generation via diffusion, which are label-guided (linear) diffusion models (Ho et al.,

2021). Note that although Ho et al. (2021) called these "classifier-free" conditional diffusion models, we will refer to them as "label-guided" in this work, since branched diffusion models also allow for conditional generation without the use of any external classifier. We trained label-guided diffusion models on the same data using the analogous architecture. We computed the Fréchet inception distance (FID) for each class, comparing branched diffusion models and their linear label-guided counterparts (Supplementary Figure S3). In general, the branched diffusion models achieved similar or better generative performance compared to the current state-of-the-art label-guided strategy. In many cases, the branched models *outperformed* the label-guided models, likely due to the multi-tasking architecture which can help limit inappropriate crosstalk between distinct classes. This establishes that branched diffusion models offer competitive performance in sample quality compared to the state-of-the-art methods for conditional diffusion.

Although we will focus our later analyses on continuous-time (i.e. SDE-based (Song et al., 2021)) diffusion models, we also trained a *discrete-time* branched model (i.e. based on DDPM (Ho et al., 2020)) to generate MNIST classes (Supplementary Figure S1). This illustrates the flexibility of branched diffusion models: as they are generally orthogonal to the underlying diffusion process, they perform equally well in both continuous- and discrete-time diffusion settings.

## 5  Extending branched diffusion models to novel classes

The problem of incorporating new data into an existing model is a major challenge in the area of continual learning, as the emergence of new classes (which were not available during training) typically requires the whole model to be retrained (van de Ven & Tolias, 2019). Conventional (linear) diffusion models are no exception, and there is a critical need to improve the extendability of these models in a continual-learning setting. This requirement is typical of models trained on large-scale, integrated scientific datasets, which grow steadily as data of new, never-before-seen classes is experimentally produced (Han et al., 2020; Almanzar et al., 2020; Lotfollahi et al., 2021). For example, large single-cell reference atlases comprising hundreds of millions of cells across organs, developmental stages, and conditions—such as the Human Cell Atlas (Regev et al., 2017)—are continuously updated as new research is published.

By separating the diffusion of different classes into distinct branches which are learned by a multi-task neural network, a branched diffusion model easily accommodates the addition of new training data (e.g. from a recent experiment). Suppose a branched model has been trained on classes $C$, and now a never-before-seen class $c'$ has been introduced. Instead of retraining the model from scratch on $C \cup \{c'\}$ for the entire diffusion timeline, a branched model can be easily extended by introducing a new branch while keeping the other branches the same (assuming $c'$ is sufficiently similar to some existing class). For example, leveraging the intrinsic structure of cell types (Han et al., 2020), a branched diffusion model can be fine-tuned on a new study—potentially including new cell types—without retraining the entire model. Formally, we extend an existing branched diffusion model by adding a new terminal branch $(s_i, t_i, C_i) = (0, t_b, \{c'\})$, where $t_b$ is determined by the algorithm in Appendix A. The new neural network has parameters $\theta = (\theta_s, \theta_0, ..., \theta_{b_{max}})$, with shared parameters $\theta_s$ and output-task-specific parameters $\theta_i$ (one for each branch). Let $\tilde{b}$ be the branch index of the new terminal branch. Then we simply need to learn $\theta_{\tilde{b}}$, training only on $c'$ for times $t \in [0, t_b]$:

$$\theta_{\tilde{b}}^* = \text{argmin}_{\theta_{\tilde{b}}} \left\{ E_{x:\text{class}(x)=c', t<t_b} \left[ \mathcal{L}(x, t, s_\theta(x, t)_{[\tilde{b}]}) \right] \right\} \tag{5}$$

To illustrate this extendability, we trained branched diffusion models on MNIST and on our large real-world RNA-seq dataset. For the MNIST experiment, we trained on three classes: 0s, 4s, and 9s. We then introduced a new class: 7s. To accommodate this new class, we added a single new branch to the diffusion model (Figure 2a). We then fine-tuned *only the newly added branch*, freezing all shared parameters and parameters for other output tasks. That is, we only trained on 7s, and only on times $t \in [s_i, t_i)$ for the newly added branch $b_i$. After fine-tuning, our branched model was capable of generating high-quality 7s *without affecting the ability to generate other digits* (Figure 2b).

In contrast, label-guided (linear) diffusion models cannot easily accommodate a new class. In our MNIST experiment, we trained a linear model on 0s, 4s, and 9s. After fine-tuning the linear model on 7s, the model

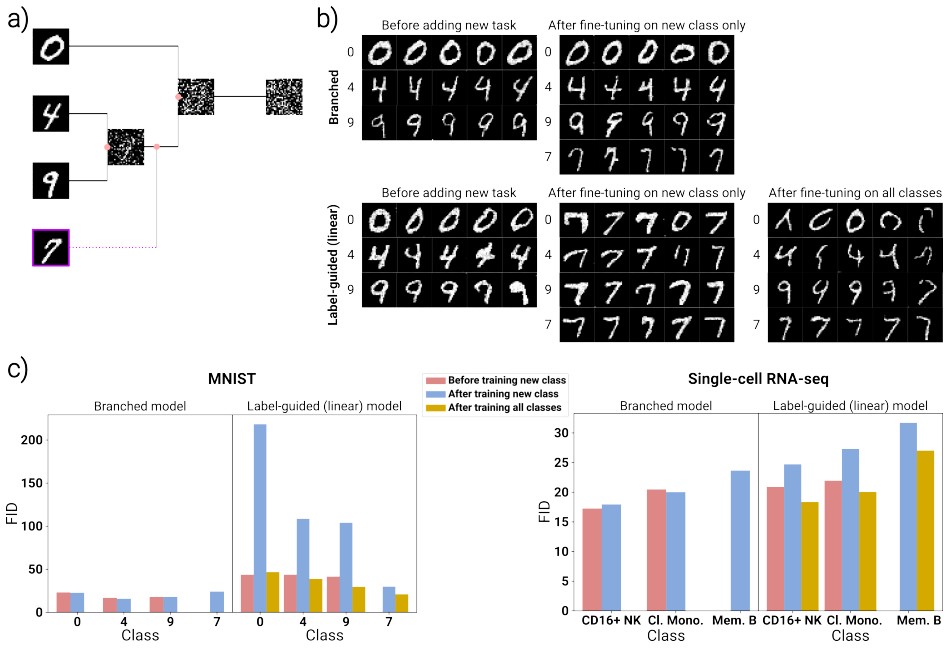

Figure 2: Extending a branched model to new classes. **a)** Schematic of the addition of a new digit class to an existing branched diffusion model on MNIST. The introduction of the new class is accomplished by adding a singular new branch (purple dotted line). **b)** Examples of MNIST digits generated from a branched diffusion model (above) and a label-guided (linear) diffusion model (below), before and after fine-tuning on the new class. For the label-guided model, we also show examples of digits after fine-tuning on the whole dataset. **c)** On the MNIST dataset (left) and the single-cell RNA-seq dataset (right), we show the FID (i.e. generative performance) of each class, before and after fine-tuning on the new class. For the label-guided models, we also show the FIDs after fine-tuning on the whole dataset.

suffered from *catastrophic forgetting*: it largely lost the ability to generate the other digits (even though their labels were fed to the model at generation time), and generated almost all 7s for any label (Figure 2b). For the linear model to retain its ability to generate pre-existing digits, it must be retrained on the *entire* dataset, which is far more inefficient, especially when the number of classes is large. In our MNIST example, retraining the linear model on all data took approximately seven times longer than training the singular new branch on a branched model. Notably, even after retraining on all data, the linear model still experienced inappropriate influence from the new task. We observed the same trend of extendability on the branched model (and inter-class interference on the linear model) in our experiments on the large real-world RNA-seq dataset.

We then *quantified* this class-extension ability by computing the FID of branched and label-guided models before and after fine-tuning (Figure 2c). On both MNIST and the RNA-seq dataset, we found that the branched models achieved roughly the same FIDs on pre-existing classes after fine-tuning on the new data class. In contrast, fine-tuning the label-guided models on the new data class caused the FID of other classes to significantly worsen. The label-guided models needed to be trained on the entire dataset to recover the FIDs of pre-existing classes, although the FIDs were still generally worse than those of the branched models.

Notably, in our extension of branched models, we were able to generate high-quality samples of the new class by *only training a single branch*. Although several upstream branches (i.e. at later diffusion times) also diffuse over the newly added class $c'$ (along with pre-existing classes), we found that fine-tuning can be performed on only the new branch which diffuses solely only $c'$, and the model still achieved high generative performance across all classes $C \cup \{c'\}$. This is a natural consequence of our explicit branch points, which are defined so that reverse diffusion at upstream branches is nearly identical between different classes. Note that even if fine-tuning were done on all classes for all $b_i$ where $c' \in C_i$, the branched diffusion model is still more efficient to extend than a linear model because most branches diffuse over only a small subset of classes in $C$.

This highlights the advantage of branched diffusion models to accommodate new classes efficiently (i.e. with little fine-tuning) and cleanly (i.e. without affecting the generation of other classes) compared to the current state-of-the-art methods for conditional diffusion. Note that although we showed the addition of a brand new class $c' \notin C$, branched models can also easily accommodate new data of an existing class $c \in C$ by fine-tuning only the appropriate branch(es).

## 6    Analogy-based conditional generation between classes

In a diffusion model, we can traverse the diffusion process both forward and in reverse. In a branched diffusion model, this allows for a unique ability to perform *analogy-based* conditional generation (or *transmutation*) between classes. That is, we start with an object of one class, and generate the analogous, corresponding object of a different class. Formally, consider the set of all branches $\{(s_i, t_i, C_i)\}$. Say we have an object $x_1$ of class $c_1$. We wish to transmute this object into the analogous object of class $c_2$. Let $t_b$ be the first branch point (earliest in diffusion time) in which $c_1$ and $c_2$ are both in the same branch. Then, in order to perform transmutation, we first forward diffuse $x_1$ to $x_b \sim q_{t_b}(x|x_1)$. Then we draw an object from the conditional distribution $p_0(x|c_2, x_b)$, where conditioning is both on class $c_2$ and the noisy object $x_b$ (partially diffused from $x_1$). In summary:

$$
\begin{aligned}
t_b &:= \min\{s_i | c_1, c_2 \in C_i\} \\
x_b &\sim q_{t_b}(x|x_1) \\
x_2 &\sim p_0(x|c_2, x_b).
\end{aligned}
\tag{6}
$$

In practice, sampling $x_2$ from $\sim p_0(x|c_2, x_b)$ is performed by reverse diffusing to generate an object of class $c_2$ (Algorithm 2), as if we started at time $t_b$ with object $x_b \sim q_{t_b}(x|x_1)$.

Conditional generation via transmutation is a unique and novel way to harness branched diffusion models for more sophisticated generation tasks which go beyond what is possible with current diffusion models, which typically condition on a single class or property (Song et al., 2021; Ho et al., 2021). In transmutation, we enable generation *conditioned on both a class and a specific instance (which may be of another class)*. That is, conventional conditional generation samples from $q_0(x|c_2)$, but transmutation samples from $q_0(x|c_2, x_1 \in c_1)$. This feature can support discovery in scientific settings. For example, given a model trained on many cell types $C$, with each cell type measured in certain conditions $h_1, ..., h_m$, transmutation can answer the following question: "*what would be the expression of a specific cell $x_i$ of cell type $c_i$ and condition $h_k$, if the cell type were $c_j$ instead?*". A branched model is thus distinct from models which simply generate cells with cell type $c_j$ and/or condition $h_k$. For instance, to study how a novel cell type (such as a B-cell) reacts to a drug whose effects are known for another cell type (such as a T-cell), one can conditionally generate a population of B-cells starting from a population of T-cells under the particular drug effect.

On our MNIST branched diffusion model, we transmuted between 4s and 9s (Figure 3a). Intriguingly, the model learned to transmute based on the *slantedness* of a digit. That is, slanted 4s tended to transmute to slanted 9s, and *vice versa*. To *quantify* analogous conditional generation between classes, we then transmuted between letters on our tabular branched diffusion model (Figure 3b). Transmuting between V and Y (and *vice versa*), we found that for every feature, there was a positive correlation of the feature values before versus after transmutation. That is to say, letters with a larger feature value tended to transmute to letters also with a larger feature value, *even if the range of the feature is different between the two classes.*

We then turned to our branched model trained on the large real-world RNA-seq dataset, and transmuted a sample of CD16+ NK cells to classical monocytes, and *vice versa*. In both directions, transmutation successfully increased critical marker genes of the target cell type, and zeroed the marker genes of the source cell type (e.g. when transmuting NK cells to monocytes, the expression of NK marker genes such as MS4A6A were zeroed, and the expression of monocyte marker genes such as SPON2 were elevated) (Figure 3c). Additionally, we found a high correlation of expression in many genes before and after transmutation, including CXCL10 ($r = 0.20$), HLA-DRA ($r = 0.16$), and HLA-DRB1 ($r = 0.15$). These genes are especially relevant, as they were explicitly featured in Lee et al. (2020) as key inflammation genes that distinguish

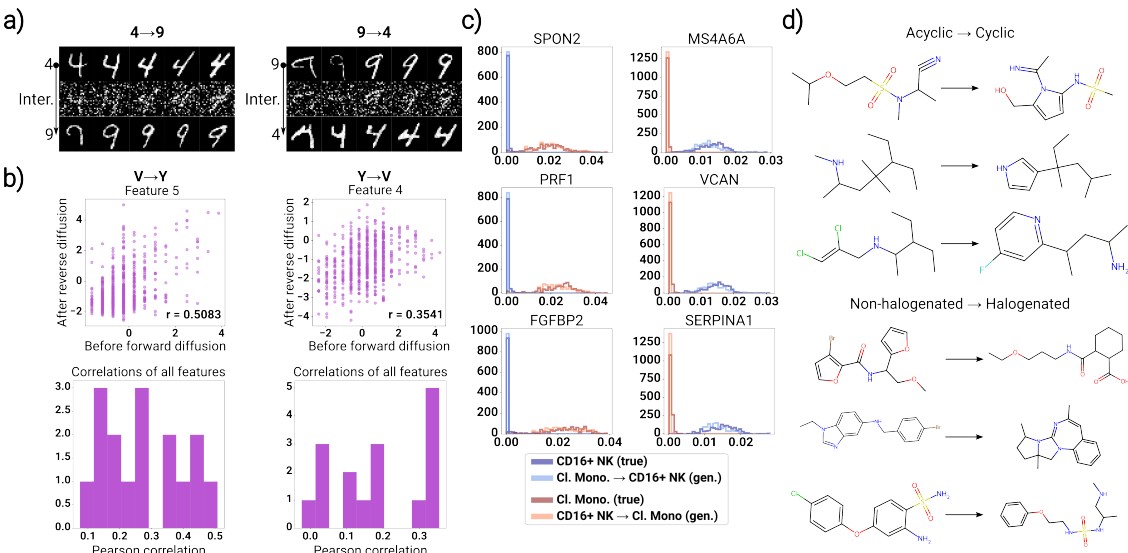

Figure 3: Transmutation between classes. **a)** From our branched diffusion model trained on MNIST, we show examples of 4s transmuted to 9s (left), and 9s transmuted to 4s (right). We also show the diffusion intermediate $x_b$ at the branch point. **b)** From our branched model trained on tabular letters, we show the scatterplots of some feature values before and after transmutation from Vs to Ys (left), or Ys to Vs (right). For each of the 16 features, we correlate the feature value before versus after transmutation and show a histogram of the correlations over all 16 features in either transmutation direction. **c)** From our branched model trained on the single-cell RNA-seq dataset, we transmuted between CD16+ NK cells and classical monocytes, and show the distribution of several marker genes before and after transmutation. The left column shows marker genes of classical monocytes, and the right column shows marker genes of CD16+ NK cells. **d)** From our branched model trained on ZINC250K, we transmuted between acyclic and cyclic molecules, and between non-halogenated and halogenated molecules.

COVID-19-infected cells from healthy cells. Notably, we recovered these strong correlations even though the original expression of these genes showed little to no distinction between healthy and infected cells (Supplementary Figure S4). This illustrates how our branched model successfully transmuted COVID-infected cells of one type into COVID-infected cells of another type (and reflexively, healthy cells from one type into healthy cells of another type).

Finally, we trained branched diffusion models on ZINC250K (Supplementary Methods), another large real-world dataset and an entirely different data modality: molecular graphs. We trained branched diffusion models to conditionally generate acyclic and cyclic molecules, or halogenated and non-halogenated molecules. We then transmuted molecules from one property class to another, while largely retaining core functional groups (e.g. amines, esters, sulfonamides, etc.) (Figure 3d). Quantitatively, transmutation from acyclic to cyclic molecules was 96.5% effective (i.e. from 0% of molecules having a cycle, we transmuted to 96.5% of molecules having a cycle). We then quantified the preservation of functional groups by computing the Jaccard index before and after transmutation. A t-test (compared to the whole set of cyclic molecules) returned $p = 7.57 \times 10^{-10}$. On the transmutation of halogenated to non-halogenated molecules, our transmutation was 100% effective, and a t-test on the preservation of functional groups returned $p = 4.55 \times 10^{-11}$.

Across our many datasets, these results together qualitatively and quantitatively show that transmutation in branched diffusion models is both: 1) *effective*—defining features of the source class are removed and defining features of the target class are generated; and 2) *analogous*—features which characterize the original object/instance (but do not directly define its class) are preserved.

# 7 Interpretability of branch-point intermediates

Interpretability is a particularly useful tool for understanding data, and is a cornerstone of AI for science. Unfortunately, there is limited work (if any) that attempts to improve or leverage diffusion-model inter-

pretability. By explicitly encoding branch points, a branched diffusion model offers unique insight into the relationship between classes and individual objects from a generative perspective.

In the forward-diffusion process of a branched diffusion model, two branches meet at a branch point when the classes become sufficiently noisy such that they cannot be distinguished from each other. Symmetrically, in the reverse-diffusion process, branch points are where distinct classes split off and begin reverse diffusing along different trajectories. Thus, for two similar classes (or two sets of classes), the reverse-diffusion intermediate at a branch point naturally encodes features which are shared (or otherwise intermediate or interpolated) between the two classes (or sets of classes).

In particular, hybrid intermediates represent partially reverse-diffused objects right before a branch splits into two distinct classes. Formally, consider the set of all branches $\{(s_i, t_i, C_i)\}$. For two classes $c_1$ and $c_2$, let $t_b$ be the first branch point (earliest in diffusion time) in which $c_1$ and $c_2$ are both in the same branch. We define a *hybrid object* $x_h$ between classes $c_1$ and $c_2$ as an object sampled from the partially diffused distribution at $t_b$ from the conditional distribution of $c_1$ or $c_2$:

$$
\begin{aligned}
t_b &:= \min\{s_i | c_1, c_2 \in C_i\} \\
x_h &\sim p_{t_b}(x|c_1) = p_{t_b}(x|c_2).
\end{aligned}
\tag{7}
$$

In practice, sampling $x_h$ from $p_{t_b}(x, c_1)$ or $p_{t_b}(x, c_2)$ is done by performing reverse diffusion following Algorithm 2 from time $T$ until time $t_b$ for either $c_1$ or $c_2$ (it does not matter which, since we stop at $t_b$).

For example, on our MNIST branched diffusion model, hybrids tend to show shared characteristics that underpin both digit distributions (Figure 4a–b). On our branched model trained on tabular letters, we see that hybrids tend to interpolate between distinct feature distributions underpinning the two classes, acting as a smooth transition state between the two endpoints (Figure 4c).

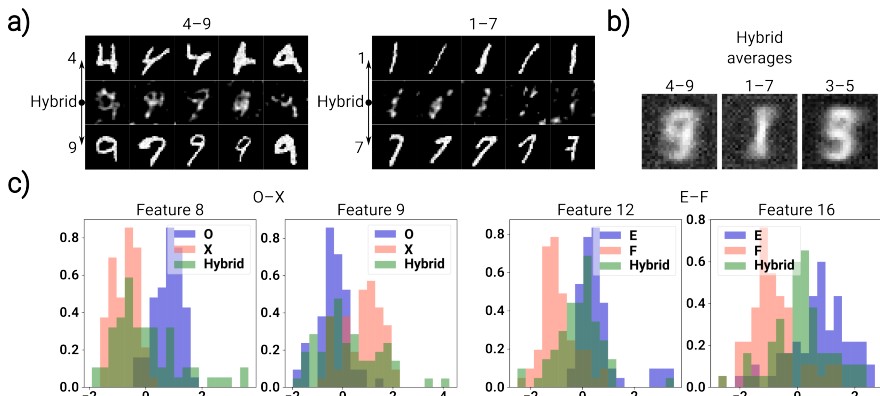

Figure 4: Interpretable hybrids at branch points. **a)** From our branched model trained on MNIST, we show examples of hybrids between the digits classes 4 and 9 (left), and between the digit classes 1 and 7 (right). Each hybrid in the middle row is the reverse-diffusion starting point for both images above and below it. We applied a small amount of Gaussian smoothing to the hybrids for ease of viewing. **b)** Averaging over many samples, the aggregate hybrids at branch points show the collective characteristics that are shared between MNIST classes. **c)** From our branched model trained on tabular letters, we show the distribution of some features between two pairs of classes—O and X (left), and E and F (right)—and the distribution of that feature in the generated hybrids from the corresponding branch point.

Note that although a branched diffusion model can successfully generate distinct classes even with very conservative branch points (i.e. late in diffusion time), the interpretability of the hybrid intermediates is best when the branch points are *minimal* (earliest in diffusion) times of indistinguishability. Taken to the extreme, branch points close to $t = T$ encode no shared information between classes whatsoever, as the distribution of objects at time $T$ is independent of class.

## 8   Efficient multi-class sampling from branched diffusion models

In addition to advantages in continual learning, transmutation, and interpretability, branched models also offer a minor benefit in generative efficiency, because partially reverse-diffused intermediates at branch points can be cached and reused. Note that branched models and standard linear models take the same amount of computation to generate a *single* class, but branched models enjoy significant savings in computational efficiency when sampling *multiple* classes (Supplementary Table S9).

## 9   Discussion

In this section, we explore some of the trade-offs and caveats of branched diffusion models.

Firstly, the multi-task neural network behind a branched diffusion model is crucial to its efficient parameterization. As previously discussed, the number of branches scales linearly with the number of classes, so the multi-task architecture is relatively efficient even for datasets with a large number of classes. Still, we recognize that an extremely large number of classes could become a bottleneck. In those cases, the branched diffusion architecture could benefit from recent advancements in efficient multi-task parameterizations (Vandenhende et al., 2022).

Additionally, the advantages and performance of branched diffusion models rely on appropriately defined branch points. We performed a robustness analysis and found that although the underlying branch points are important, *the performance of branched diffusion models is robust to moderate variations in these branch points* (Figure S5). Our branch-point discovery algorithm (Appendix A) is also agnostic to the diffusion process, and although it relies on Euclidean distance between noisy objects (which may be hard to compute for data types like graphs), the algorithm (and subsequent diffusion) can always be applied in latent space to guarantee well-defined Euclidean distances.

Finally, branched diffusion models may have difficulties learning on certain image datasets where the class-defining subject of the image can be in different parts of the image, particularly when data may be sparse. For datasets like MNIST, the digits are all roughly in the center of the image, thus obviating this problem. Of course, images and image-like data are the only modalities that suffer from this issue. Additionally, this limitation on images may be avoided by diffusing in latent space.

## 10   Conclusion

In this work, we proposed a novel form of diffusion models which introduces branch points which explicitly encode the hierarchical relationship between distinct data classes. Branched diffusion models are an alternative method of conditional generation for discrete classes. Compared to the current state-of-the-art conditional diffusion models, we showed numerous advantages of branched models in conditional generation. We showcased these advantages across many different datasets, including several standard benchmark datasets and two large real-world scientific datasets.

Firstly, we showed that branched models are easily extendable to new, never-before-seen classes through an efficient fine-tuning step which does not lead to catastrophic forgetting of pre-existing classes. This can enhance diffusion-model training in online-learning settings and in scientific applications where new data is constantly being produced experimentally. Additionally, branched models are capable of more sophisticated forms of conditional generation, such as the transmutation of objects from one class into the analogous object of another class. Using transmutation, we demonstrated the ability of branched diffusion models to discover relevant biology and chemistry. Furthermore, we showed that branched models can offer some insights into interpretability. Namely, reverse-diffusion intermediates at branch points are hybrids which encode shared or interpolated characteristics of multiple data classes.

Finally, because branched diffusion models operate on the same underlying diffusion process as a conventional linear model, they are flexibly applied to virtually any diffusion-model paradigm (e.g. continuous or discrete time, SDE based or Markov-chain based, different noise schedules and definitions of the noising process, etc.). Branched models are also easily combined with existing methods which aim to improve training/sampling

efficiency or generative performance (Kong & Ping, 2021; Watson et al., 2021; Dockhorn et al., 2021; Song et al., 2021; Xiao et al., 2022), or other methods which condition based on external properties (Song et al., 2021; Ho et al., 2021).

Branched diffusion models have many direct applications, and we highlighted their usefulness in scientific settings. Further exploration in the structure of diffusion models (e.g. branched vs linear) may continue to have resounding impacts in how these models are used across many areas.

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

# A   Branch-point discovery

As objects from two different classes are forward diffused, their distributions become more and more similar to each other. At the extreme, it is expected that all objects (regardless of class) reach the same prior distribution at the time horizon $T$. Our goal is quantify the earliest time point when objects of two different classes are sufficiently *similar* in distribution, such that the reverse diffusion after that point can be predicted by the same model between the two classes (without specifying class identity).

More formally, we define a branch point between two classes $c_1$ and $c_2$ as the diffusion time $t_b$ where the objects sampled from these two classes are "relatively indistinguishable" from each other. Because the data distribution is typically complex and high dimensional, we quantify indistinguishability as log-fold change of expected Euclidean distances, comparing data drawn between the two different classes, and data within the same class. That is, the branch point $t_b$ between classes $c_1$ and $c_2$ is defined as the *minimum* time $t$ such that:

$$\mathbb{E}_{a \in \mathcal{D}_1, b \in \mathcal{D}_2, c \in \mathcal{D}_2} \left[ \mathbb{E}_{x_a \sim q_t(x|a), x_b \sim q_t(x|b), x_c \sim q_t(x|c)} \left[ \log(\frac{\|x_a - x_b\|_2}{\|x_c - x_b\|_2}) \right] \right] < \epsilon \tag{8}$$

where $\mathcal{D}_i$ is the distribution of data with class label $c_i$.

We discuss theoretical justifications and connections in Appendix B.

The outer expectations are taken over points in the dataset, and the inner expectations are over the forward noising process. When this log-fold change is a small number $\epsilon$, then we consider the classes relatively indistinguishable (i.e. the average distance between noisy objects of different classes is comparable to the average distance between noisy objects of the same class).

Of course, these expectations are extremely intractable to compute in closed-form because of the large dataset and dimensionality, so instead we approximate these quantities by using Monte Carlo sampling. That is, we take a sample of objects from each class, apply forward diffusion at many times $t$ spaced regularly between 0 and $T$, and identify the smallest $t$ such that log-fold change of the distance is sufficiently small.

This procedure gives a branch point $t_b$ for every pair of classes $c_i, c_j$ ($i \neq j$).

In a branched diffusion model, each branch $b_i = (s_i, t_i, C_i)$ learns to reverse diffuse between times $[s_i, t_i)$ for classes in $C_i$. The branches form a tree structure (i.e. hierarchy) with the root at time $T$ and a branch for each individual class at time 0. In order to convert the branch points between all pairs of classes into a hierarchy, we simply perform a greedy aggregation starting from individual classes and iteratively merge classes together by their branch points (from early to late diffusion time) until all classes have been merged together.

To summarize, the full branch-point discovery algorithm is as follows:

1. Start with a dataset of objects to generate, consisting of classes $C$.

2. For each class, sample $n$ objects randomly and without replacement.

3. Forward diffuse each object over 1000 time points in the forward-diffusion process (we used 1000 steps, as this matched the number of reverse-diffusion steps we used for sample generation). The branched diffusion model which will be trained using these branch definitions employs an identical forward-diffusion process.

4. At each time point $t$, compute the average Euclidean distance of each pair of classes, resulting in a $|C| \times |C|$ matrix at each of the 1000 time points. For distinct classes $c_i, c_j$ ($i \neq j$), the distance $s(t, c_i, c_j)$ is computed over the average of $n$ pairs, where the pairs are randomly assigned between the two classes; for self-distance of class $c_i$, the distance $s(t, c_i, c_i)$ is computed over the average of $n$ pairs within the class, randomly assigned such that the same object is not compared with itself.

5. For each pair of classes $c_i, c_j$ ($i$ may be equal to $j$), smooth the trajectory of $s(t, c_i, c_j)$ over time by applying a Gaussian smoothing kernel of standard deviation equal to 3 and truncated to 4 standard deviations on each side.

6. For each pair of *distinct* classes $c_i, c_j$ $(i \neq j)$, compute the *earliest* time in the forward-diffusion process such that the log-fold change of the average distance between $c_i$ and $c_j$ over the self-distance of $c_i$ and $c_j$ (averaged between the two) is at most a tolerance of $\epsilon$. That is, for each pair of distinct classes $c_i, c_j$ $(i \neq j)$, compute the *minimum t* such that $\log(\frac{s(t,c_i,c_j)}{\frac{1}{2}(s(t,c_i,c_i)+s(t,c_j,c_j))}) < \epsilon$. This gives each pair of distinct classes a "minimal time of indistinguishability", $\tau_{c_i,c_j}$.

7. Order the $\binom{|C|}{2}$ minimal times of indistinguishability $\tau$ by ascending order, and greedily build a hierarchical tree by merging classes together if they have not already been merged. This can be implemented by a set of $|C|$ disjoint sets, where each set contains one class; iterating through the times $\tau$ in order, two branches merge into a new branch by merging together the sets containing the two classes, unless they are already in the same set.

## B   Theoretical justification for branch-point-discovery algorithm

The goal of our branch-point discovery algorithm is to find the branch point between any pair of classes $c_1$ and $c_2$. The branch point is the earliest point in diffusion time where $c_1$ and $c_2$ are "indistinguishable" from each other. Formally, indistinguishability occurs when the conditional distributions of these two classes are close enough that they can be modeled by a single reverse-diffusion process. That is, we want the branch point $t_b$ to be such that $q_{t_b}(x|c_1) \approx q_{t_b}(x|c_2)$.

Of course, this is a difficult condition to formally define and satisfy, and there are several challenges:

1. We only have access to samples of objects from each class' conditional distribution.

2. The data can be high dimensional, and particularly because we only have limited samples, the curse of dimensionality emerges quickly, even with as few as 10 dimensions.

3. These noisy conditional distributions $q_t(x|c)$ are not tractable to compute or even represent, particularly at earlier diffusion times (when the distribution $q_t(x)$ is so close to the unknown and complex data manifold $q_0(x)$).

4. The noisy conditional distributions for two different classes will generally not be identical (until the time horizon $T$), and they slowly approach indistinguishability asymptotically.

5. In order to find the optimal $t_b$, we must compare $q_t(x|c_1)$ and $q_t(x|c_2)$ at potentially all time points along the diffusion timeline, as well as across all pairs of classes in the dataset, so this comparison needs to be extremely computationally efficient.

We designed our branch-point-discovery algorithm based on these challenges and desiderata, leading to our definition of a branch point in Equation 8.

Notably, Equation 8 is closely related to *energy distance* (Székely & Rizzo, 2013). Energy distance compares the expected distance between objects of two different distributions, to the average expected distance within the same distribution:

$$E(\mathcal{D}_1, \mathcal{D}_2) = 2\mathbb{E}_{a \in \mathcal{D}_1, b \in \mathcal{D}_2}[\|a - b\|_2] - \mathbb{E}_{a \in \mathcal{D}_1, b \in \mathcal{D}_1}[\|a - b\|_2] - \mathbb{E}_{a \in \mathcal{D}_2, b \in \mathcal{D}_2}[\|a - b\|_2]$$

We chose to use energy distance for several reasons:

- Energy distance (by definition) compares the expected distance between two different distributions to the expected distance *within* the distributions. This allows us to attain a measure of "relative indistinguishability". By picking a time point when the energy distance is low, we ensure that the model upstream of the branch point has no more difficulty representing and learning both classes together, as it would to learn only a single class (as in a standard, linear diffusion model). This addresses Challenge 4.

- Energy distance is much more robust in high dimensions. In high dimensions, particularly with sparser samples, metrics such as MMD and significance testing such as t-tests can be unreliable (Ramdas et al., 2014; Bischoff et al., 2024). This addresses Challenges 1 and 2.

- Energy distance does not require any distributional assumptions. Furthermore, we modify the energy distance formulation slightly, so that instead of taking the *difference* between expected inter-distribution and intra-distribution distances, we take the *ratio*, as this allows for the energy distance to be more comparable in magnitude across different classes and times, even as the distributions change drastically across diffusion time. It also allows us to select $\epsilon$ in a more principled and comparative manner across different datasets. In contrast, most frameworks for significance testing relies on strong distributional assumptions, particularly in high dimensions. This addresses Challenge 3.

- Energy distance is also extremely efficient to compute. It involves computing expected distance between objects of the two distributions, as well as between objects of the same distribution. In our algorithm, we perform Monte Carlo sampling over possible random couplings between objects to compute these expected distances. This make energy distance far more suitable for our needs compared to other metrics such as Wasserstein distance, which requires minimizing over all possible couplings. This addresses Challenge 5.

- Energy distance is calculated simply using Euclidean distance, and there is intuitive reason for why Euclidean distance in diffusion space is a meaningful measurement of distance between distributions. The Euclidean distance can measure how much "work" a diffusion model does to push objects from one point to another. In particular, we can view the reverse-diffusion process as taking a sampled object from $\pi(x)$ and pushing the feature values to a final generated sample. For a single branch representing multiple classes over some diffusion time interval, distances between objects reflect the expected distance the branch will be pushing objects, and we would like this distance to be no more for objects of different classes versus objects of the same class in that branch.

## C   Supplementary Figures and Tables

Continuous-time diffusion model                    Discrete-time diffusion model

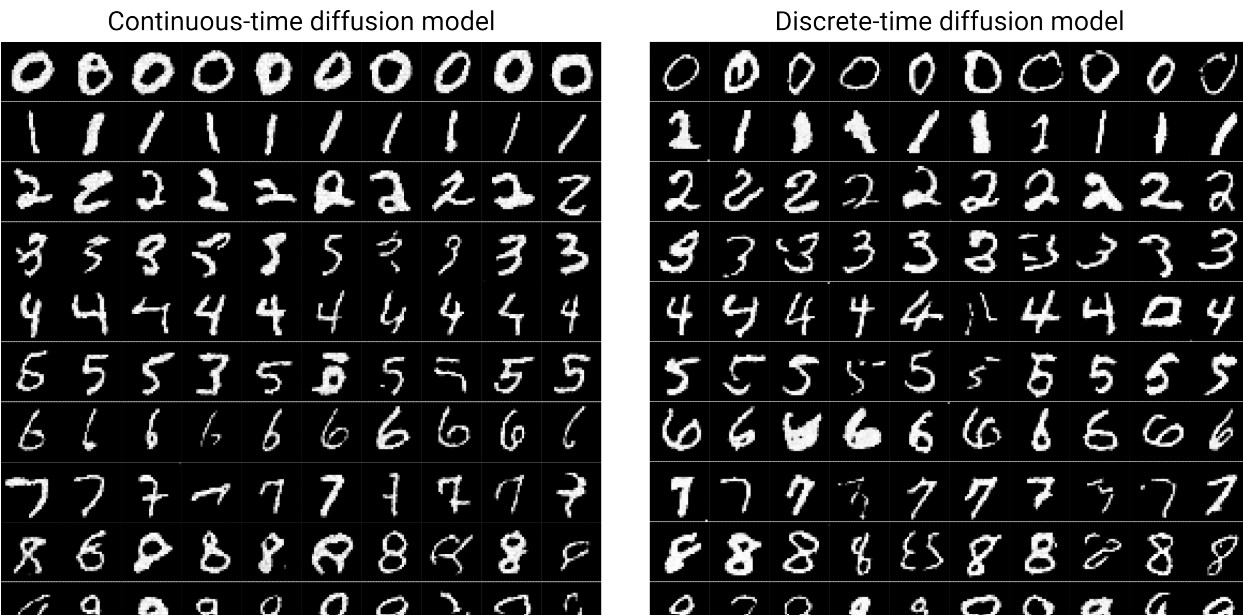

Figure S1: Examples of generated MNIST images. We show (uncurated) images of MNIST digits generated by branched diffusion models. Since branched diffusion models naturally output each class separately, generation of individual classes does not require supplying labels or pretrained classifiers. We show a sample of digits generated from a continuous-time (score-matching) diffusion model (Song et al., 2021), and a discrete-time diffusion model (denoising diffusion probabilistic model) (Ho et al., 2020). Branched diffusion models for multi-class generation fit neatly into practically any diffusion-model framework.

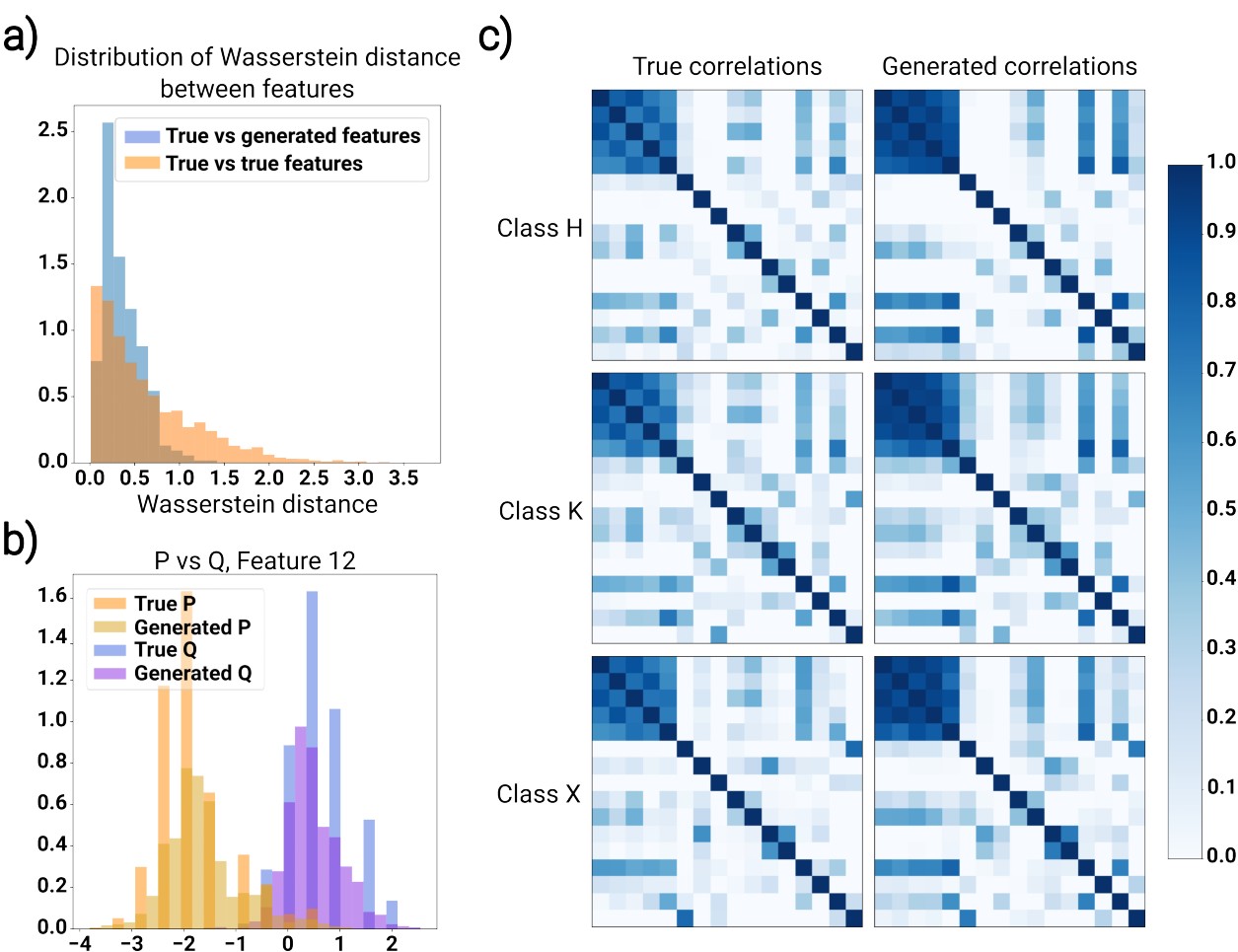

Figure S2: **Examples of generated letters.** We show some examples of distributions generated from a branched diffusion model trained on tabular data: English letters of various fonts, featurized by a hand-engineered set of 16 features (Frey & Slate, 1991). **a)** For each letter class and each of the 16 numerical features, we computed the Wasserstein distance (i.e. earthmover's distance) between the true data distribution and the generated data distribution. We compare this distribution of Wasserstein distances to the distribution of Wasserstein distances between different true features as a baseline. On average, the branched diffusion model learned to generate features which are similar in distribution to the true data. **b)** We show an example of the true and generated feature distributions for a particular feature, comparing two letter classes: P and Q. Although the two classes show a very distinct distribution for this feature, the branched diffusion model captured this distinction well and correctly generated the feature distribution for each class. **c)** Over all 16 numerical features, we computed the Pearson correlation between the features, and compared the correlation heatmaps between the true data and the generated examples. In each of these three classes, the branched diffusion model learned to capture not only the overarching correlational structure shared by all three classes, but also the subtle secondary correlations unique to each class.

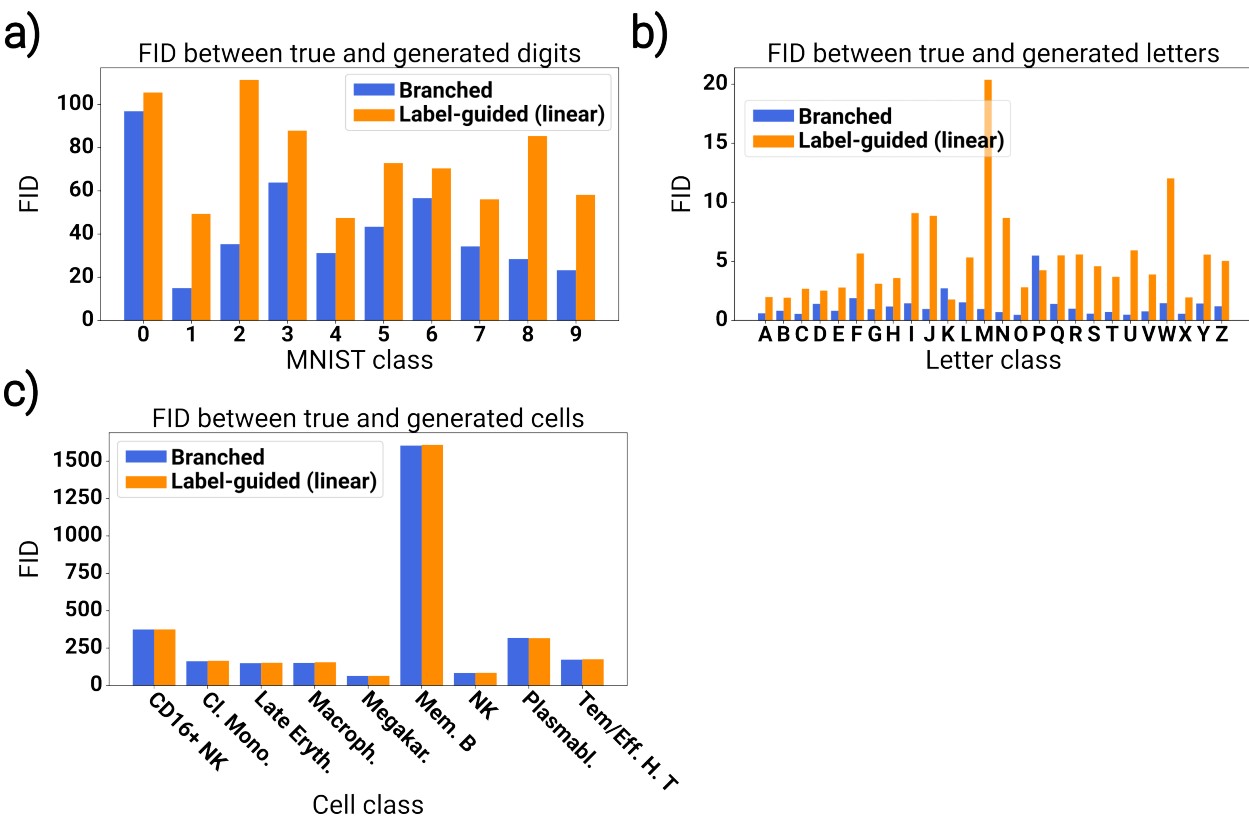

Figure S3: Sample quality of branched diffusion vs label-guided (linear) diffusion. We compare the quality of generated data from branched diffusion models to label-guided (linear) diffusion models of similar capacity and architecture. For each class, we computed the Fréchet inception distance (FID) between the generated examples and a sample of the true data. A lower FID is better. We show the FID for generated **a)** MNIST digits; **b)** tabular letters; and **c)** single-cell RNA-seq. We find that our branched diffusion model achieved comparable sample quality compared to the current state-of-the-art method of label-guided diffusion. In some cases, the branched model even consistently generated better examples.

Table S1: MNIST branch definitions

| Branch start $(s_i)$ | Branch end $(t_i)$ | Branch classes $(C_i)$ |
|---|---|---|
| 0.4855 | 1 | 0,1,2,3,4,5,6,7,8,9 |
| 0.4474 | 0.4855 | 1,2,3,4,5,6,7,8,9 |
| 0.4334 | 0.4474 | 2,3,4,5,6,7,8,9 |
| 0.4164 | 0.4334 | 2,3,4,5,7,8,9 |
| 0.3744 | 0.4164 | 3,4,5,7,8,9 |
| 0.3684 | 0.3744 | 3,4,5,8,9 |
| 0.3524 | 0.3684 | 3,4,5,9 |
| 0.3483 | 0.3524 | 3,4,5 |
| 0.2713 | 0.3483 | 3,5 |
| 0 | 0.4855 | 0 |
| 0 | 0.4474 | 1 |
| 0 | 0.4334 | 6 |
| 0 | 0.4164 | 2 |
| 0 | 0.3744 | 7 |
| 0 | 0.3684 | 8 |
| 0 | 0.3524 | 9 |
| 0 | 0.3483 | 4 |
| 0 | 0.2713 | 5 |
| 0 | 0.2713 | 3 |

Branch definitions for model on all MNIST digits.

Table S2: MNIST (discrete) branch definitions

| Branch start $(s_i)$ | Branch end $(t_i)$ | Branch classes $(C_i)$ |
|---|---|---|
| 761 | 1000 | 0,1,2,3,4,5,6,7,8,9 |
| 760 | 761 | 0,2,3,4,5,6,7,8,9 |
| 712 | 760 | 2,3,4,5,6,7,8,9 |
| 709 | 712 | 3,4,5,6,7,8,9 |
| 700 | 709 | 3,5,6,8 |
| 685 | 709 | 4,7,9 |
| 659 | 700 | 3,5,8 |
| 656 | 659 | 3,5 |
| 527 | 685 | 4,9 |
| 0 | 761 | 1 |
| 0 | 760 | 0 |
| 0 | 712 | 2 |
| 0 | 700 | 6 |
| 0 | 685 | 7 |
| 0 | 659 | 8 |
| 0 | 656 | 5 |
| 0 | 656 | 3 |
| 0 | 527 | 4 |
| 0 | 527 | 9 |

Branch definitions for discrete-time model on all MNIST digits.

Table S3: MNIST branch definitions for 0, 4, 9

| Branch start $(s_i)$ | Branch end $(t_i)$ | Branch classes $(C_i)$ |
|---|---|---|
| 0.5 | 1 | 0,4,9 |
| 0 | 0.5 | 0 |
| 0.35 | 0.5 | 4,9 |
| 0 | 0.35 | 4 |
| 0 | 0.35 | 9 |

Branch definitions for model on MNIST digits 0, 4, and 9.

Table S4: MNIST branch definitions for 0, 4, 7, and 9

| Branch start $(s_i)$ | Branch end $(t_i)$ | Branch classes $(C_i)$ |
|---|---|---|
| 0.5 | 1 | 0,4,7,9 |
| 0 | 0.5 | 0 |
| 0.38 | 0.5 | 4,7,9 |
| 0 | 0.38 | 7 |
| 0.35 | 0.38 | 4,9 |
| 0 | 0.35 | 4 |
| 0 | 0.35 | 9 |

Branch definitions for model on MNIST digits 0, 4, 7, and 9.

Table S5: Letters branch definitions

| Branch start $(s_i)$ | Branch end $(t_i)$ | Branch classes $(C_i)$ |
|---|---|---|
| 0.5235 | 1 | A,B,C,D,E,F,G,H,I,J,K,L,M,N,O,P,Q,R,S,T,U,V,W,X,Y,Z |
| 0.5165 | 0.5235 | A,B,C,D,E,F,G,H,I,J,K,L,M,N,O,P,Q,R,S,T,U,V,X,Y,Z |
| 0.5115 | 0.5165 | B,C,D,E,F,G,H,I,J,K,L,M,N,O,P,Q,R,S,T,U,V,X,Y,Z |
| 0.4945 | 0.5115 | B,C,D,E,F,G,H,I,J,K,M,N,O,P,Q,R,S,T,U,V,X,Y,Z |
| 0.4795 | 0.4945 | I,J |
| 0.4725 | 0.4945 | B,C,D,E,F,G,H,K,M,N,O,P,Q,R,S,T,U,V,X,Y,Z |
| 0.4565 | 0.4725 | B,C,D,E,G,H,K,M,N,O,Q,R,S,U,X,Z |
| 0.4364 | 0.4725 | F,P,T,V,Y |
| 0.4174 | 0.4565 | B,C,D,E,G,H,K,N,O,Q,R,S,U,X,Z |
| 0.4134 | 0.4174 | B,C,D,E,G,H,K,N,O,Q,R,S,X,Z |
| 0.4094 | 0.4134 | B,D,G,H,K,N,O,Q,R,S,X,Z |
| 0.4024 | 0.4364 | F,T,V,Y |
| 0.3864 | 0.4094 | B,D,G,H,K,O,Q,R,S,X,Z |
| 0.3814 | 0.3864 | B,G,H,K,O,Q,R,S,X,Z |
| 0.3734 | 0.3814 | B,G,H,O,Q,R,S,X,Z |
| 0.3604 | 0.4024 | F,T,Y |
| 0.3564 | 0.4134 | C,E |
| 0.3534 | 0.3604 | T,Y |
| 0.3514 | 0.3734 | B,R,S,X,Z |
| 0.3413 | 0.3734 | G,H,O,Q |
| 0.3223 | 0.3514 | B,S,X,Z |
| 0.2763 | 0.3223 | B,S,X |
| 0.2643 | 0.3413 | G,H,O |
| 0.2573 | 0.2643 | G,O |
| 0.1562 | 0.2763 | S,X |
| 0 | 0.5235 | W |
| 0 | 0.5165 | A |
| 0 | 0.5115 | L |
| 0 | 0.4795 | J |
| 0 | 0.4795 | I |
| 0 | 0.4565 | M |
| 0 | 0.4364 | P |
| 0 | 0.4174 | U |
| 0 | 0.4094 | N |
| 0 | 0.4024 | V |
| 0 | 0.3864 | D |
| 0 | 0.3814 | K |
| 0 | 0.3604 | F |
| 0 | 0.3564 | E |
| 0 | 0.3564 | C |
| 0 | 0.3534 | Y |
| 0 | 0.3534 | T |
| 0 | 0.3514 | R |
| 0 | 0.3413 | Q |
| 0 | 0.3223 | Z |
| 0 | 0.2763 | B |
| 0 | 0.2643 | H |
| 0 | 0.2573 | G |
| 0 | 0.2573 | O |
| 0 | 0.1562 | S |
| 0 | 0.1562 | X |

Branch definitions for model on all letters.

Table S6: Single-cell RNA-seq branch definitions

| Branch start $(s_i)$ | Branch end $(t_i)$ | Branch classes $(C_i)$ |
|---|---|---|
| 0.6436 | 1 | CD16+ NK, Cl. Mono., Late Eryth., Macroph., Megakar., Mem. B, NK, Plasmabl., Tem/Eff. H. T |
| 0.5405 | 0.6436 | CD16+ NK, Cl. Mono., Late Eryth., Macroph., Megakar., NK, Plasmabl., Tem/Eff. H. T |
| 0.5085 | 0.5405 | Cl. Mono., Late Eryth., Macroph., Megakar., NK, Tem/Eff. H. T |
| 0.4505 | 0.5405 | CD16+ NK, Plasmabl. |
| 0.3724 | 0.5085 | Cl. Mono., Late Eryth., Megakar., NK, Tem/Eff. H. T |
| 0.3644 | 0.3724 | Megakar., NK, Tem/Eff. H. T |
| 0.2292 | 0.3724 | Cl. Mono., Late Eryth. |
| 0.1842 | 0.3644 | Megakar., NK |
| 0 | 0.6436 | Mem. B |
| 0 | 0.5085 | Macroph. |
| 0 | 0.4505 | CD16+ NK |
| 0 | 0.4505 | Plasmabl. |
| 0 | 0.3644 | Tem/Eff. H. T |
| 0 | 0.2292 | Cl. Mono. |
| 0 | 0.2292 | Late Eryth. |
| 0 | 0.1842 | NK |
| 0 | 0.1842 | Megakar. |

Branch definitions for model on all single-cell RNA-seq cell types.

Table S7: Single-cell RNA-seq branch definitions for CD16+ NK and Classical Monocytes

| Branch start $(s_i)$ | Branch end $(t_i)$ | Branch classes $(C_i)$ |
|---|---|---|
| 0.5796 | 1 | CD16+ NK, Cl. Mono. |
| 0 | 0.5796 | CD16+ NK |
| 0 | 0.5796 | Cl. Mono. |

Branch definitions for model on all single-cell RNA-seq cell types CD16+ NK and Classical Monocytes

Table S8: Single-cell RNA-seq branch definitions for CD16+ NK, Classical Monocytes, and Memory B

| Branch start $(s_i)$ | Branch end $(t_i)$ | Branch classes $(C_i)$ |
|---|---|---|
| 0.6787 | 1 | CD16+ NK, Cl. Mono., Mem. B |
| 0.5796 | 0.6787 | CD16+ NK, Cl. Mono. |
| 0 | 0.6787 | Mem. B |
| 0 | 0.5796 | CD16+ NK |
| 0 | 0.5796 | Cl. Mono. |

Branch definitions for model on all single-cell RNA-seq cell types CD16+ NK, Classical Monocytes, and Memory B

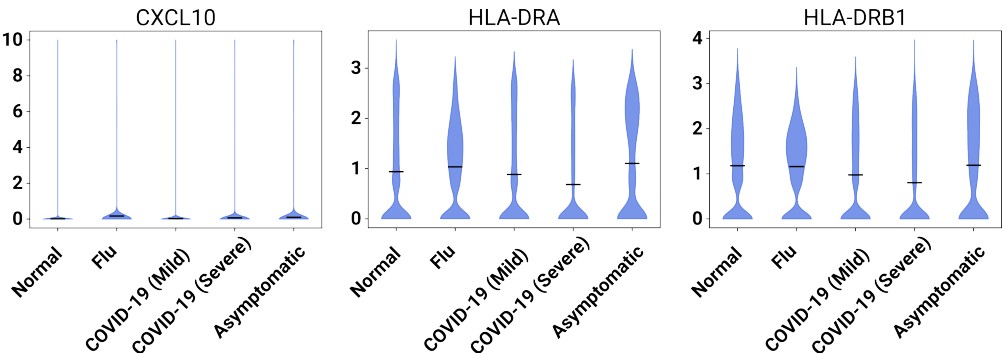

Figure S4: True expression of genes by cell population. We show the distribution of true (normalized) expression of CXCL10, HLA-DRA, and HLA-DRB1 across different cell populations (i.e. healthy/normal, COVID-19, etc.). The average expression in each population is denoted by a black line. Due to complex interactions between genes and the highly noisy measurements that are characteristic of single-cell RNA-seq, the distribution of the expression of these genes generally shows no trivial differences in cell populations infected with COVID-19 relative to healthy cells. However, important differences in the distribution of these genes between COVID-19 and healthy cell populations can be recovered through more advanced computational methods, as detailed in (Lee et al., 2020).

Table S9: Efficiency of multi-class conditional generation

| Dataset | Linear model | Branched model |
|---|---|---|
| MNIST | $78.73 \pm 0.11$ | $\mathbf{37.30} \pm 0.03$ |
| Letters | $110.42 \pm 0.14$ | $\mathbf{67.54} \pm 0.04$ |
| Single-cell RNA-seq | $275.81 \pm 0.02$ | $\mathbf{132.37} \pm 0.01$ |

When generating data from multiple classes, intermediates at branch points can be cached in a branched diffusion model. For three datasets, we measure the time taken to generate one batch of each class from a branched diffusion model, and from a label-guided (linear) model of identical capacity. Averages and standard errors are shown over 10 trials each. All values are reported as seconds.

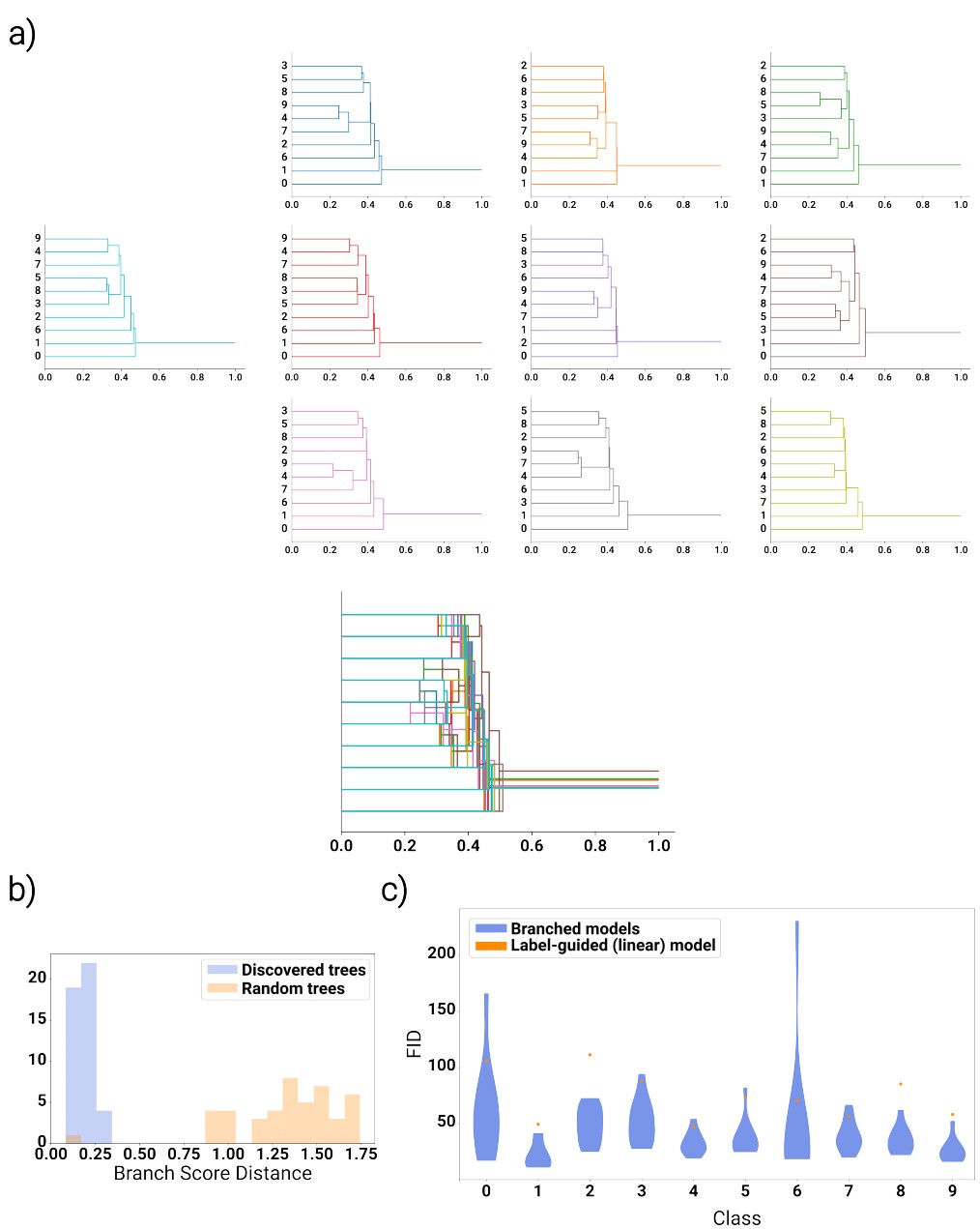

Figure S5: Robustness of branch points. **a)** We computed branch points and hierarchies for the MNIST dataset 10 times, each time resulting in a slightly different branching structure. The variation results from randomness in sampling from the dataset, and stochasticity in the forward-diffusion process. The 10 branching structures vary not only in their branching times, but also in their topologies (above). To emphasize the variation in the hierarchies, we also overlay all 10 hierarchies on the same axes (below). **b)** Compared to randomly generated hierarchies, the branching structures generated by our algorithm (Appendix A) have a much lower branch-score distance between themselves ($p < 10^{-25}$ by Wilcoxon test). **c)** We trained a branched diffusion model on each of the hierarchies, and quantified generative performance using Fréchet inception distance (FID). Over all 10 hierarchies, the FID from the branched models were relatively consistent with each other, and also generally better than the label-guided (linear) model.

Table S10: Relative size and capacity of branched models vs relative training time

| Branched model | Number of classes | Total branch length | Model capacity | Actual epochs |
|---|---|---|---|---|
| MNIST | 10 | 11.49 | 8.48 | 3 |
| Fashion MNIST | 10 | 4.81 | 8.48 | 2 |
| Letters | 26 | 10.05 | 11.06 | 1 |
| Single-cell RNA-seq | 9 | 4.29 | 6.38 | 1.2 |
| ZINC250K | 2 | 1.15 | 1.89 | 1 |

Because branched diffusion models separate the diffusion process into branches, they naturally often require more epochs to train than their linear counterparts. We found, however, that the time taken to train these models was still far less than what would be expected if the training time were an additive function of total branch length or model capacity. Here, we show (for each of our main branched models) the total branch length (i.e. the sum of diffusion times over all branches $b_i$: $\sum_{i=1}^{2|C|-1} (t_i - s_i)$), the approximate model capacity, and the actual training time in epochs. All values in the table above are *relative* (i.e. divided by) the branched model's linear counterpart. Although training time for branched models was already far less than what is expected based on branch length or model capacity, we also suggest that training with larger batch sizes or accumulating gradient updates between batches (particularly for shorter branches) may also allow for these models to be trained even faster. We leave the exploration of this for future work.

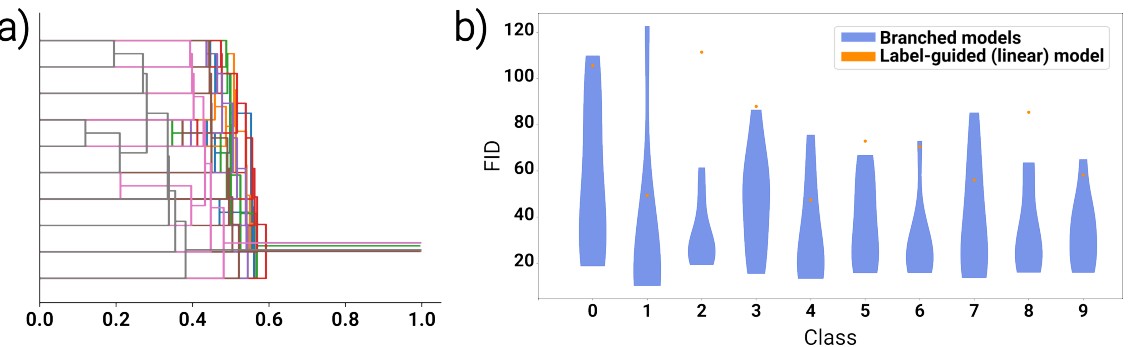

Figure S6: Robustness of $\epsilon$ in branch-point discovery algorithm. The value $\epsilon$ is used in the branch-point discovery algorithm (Appendix A) to determine when two classes are sufficiently similar to be combined in a branch. Although $\epsilon$ is relatively easy to select by simply choosing a value where the branch points are not all too close to $t = 0$ or $t = T$, here we explore the robustness of branched diffusion models to different choices of $\epsilon$. **a)** We sampled 10 values of $\epsilon$ between $10^{-5}$ and $10^{-1}$ (uniformly sampled in logarithmic space), and computed branch points for each in our MNIST dataset. The two largest values of $\epsilon$ yielded hierarchies where the terminal branches were all length 0, so they were removed from this analysis. We show an overlay of the hierarchies. Note the similarity of the hierarchies here (which arise from different values of $\epsilon$) compared to the distribution of hierarchies in Supplementary Figure S5a (which arise from random variation in sampling and forward diffusion from the same value of $\epsilon$). **b)** We trained a branched diffusion model on each of the hierarchies, and quantified generative performance using Fréchet inception distance (FID). Over all hierarchies, the FID from the branched models were mostly consistent with each other, and also mostly better than the label-guided (linear) model. The model which performed the worst arose from the largest value of $\epsilon$ in the analysis, which resulted in the hierarchy with the shortest terminal branches (gray hierarchy in panel **a**).

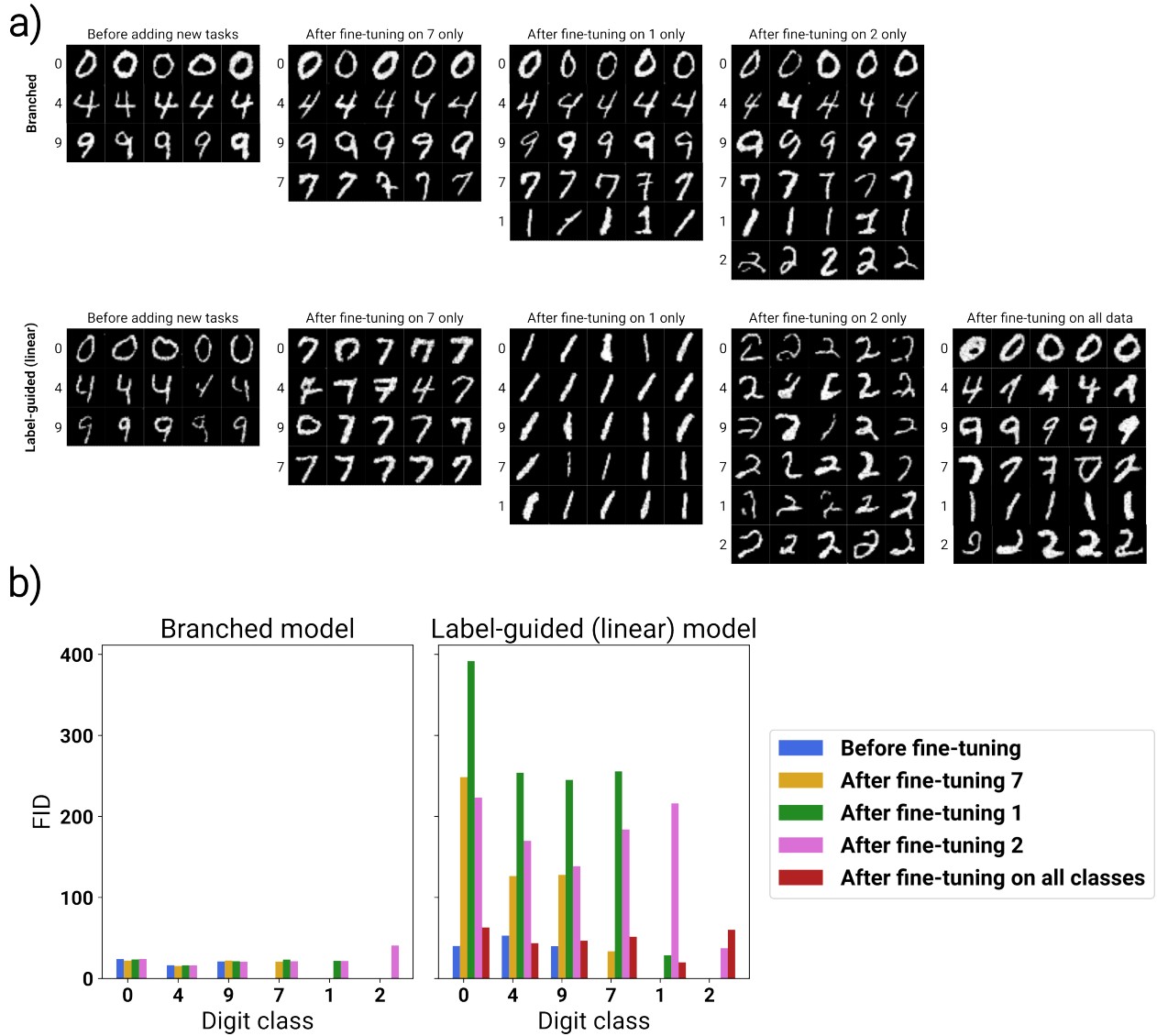

Figure S7: Multi-step continual learning. We continue the same analysis as shown in Figure 5, but show a multi-step continual-learning scheme, in which 3 never-before-seen classes are iteratively introduced to a model. We started with a branched diffusion model trained to generate 0s, 4s, and 9s. We then iteratively introduced 7s, 1s, and 2s. With the introduction of each new digit class, we added a single new terminal branch, using the branch point computed using the branch-point discovery algorithm (Appendix A). We then fine-tuned only that branch for only that class. For the label-guided model, we also began with a model trained to generate 0s, 4s, and 9s. We then iteratively introduced 7s, 1s, and 2s. With the introduction of each new digit class, we fine-tuned the model on only that digit class. We also attempted fine-tuning the label-guided model by training on all digits (old and new). **a)** We show examples of digits generated after each fine-tuning step. The branched model was able to generate high-quality digits upon the addition of each new digit class, without affecting the ability to generate pre-existing classes. The label-guided model, however, suffered from catastrophic forgetting upon being fine-tuned at each step, and largely forgot how to generate all other digits. Upon fine-tuning on all digit classes (which is a highly inefficient procedure), the label-guided model was able to generate all classes once more, but still suffered from inappropriate crosstalk between the classes. **b)** We quantified the generative performance of each model using Fréchet inception distance (FID). The branched models achieved roughly the same FIDs upon fine-tuning on each new data class, whereas the FID of the label-guided models suffered enormously as a result of catastrophic forgetting.

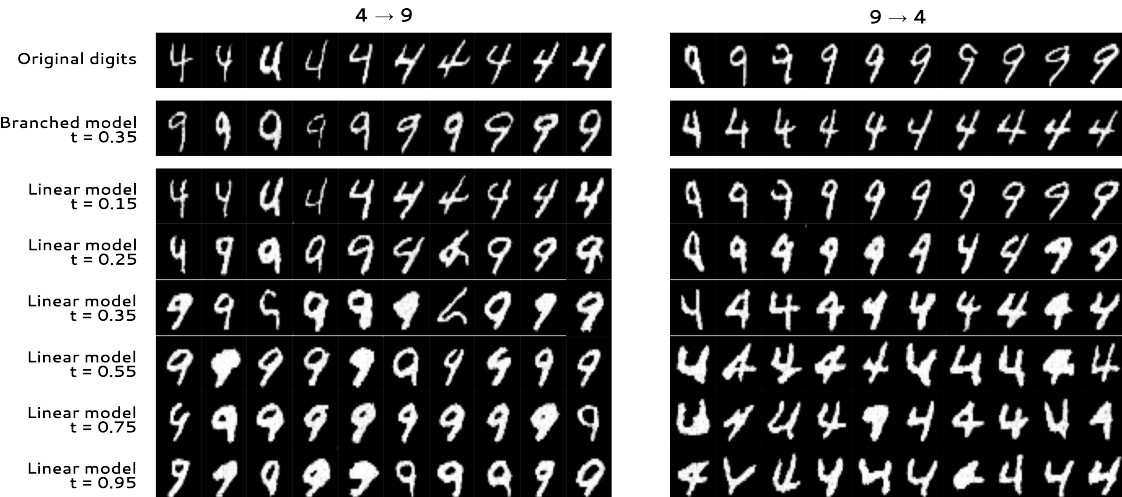

Figure S8: Transmutation in a (label-guided) linear model. Transmutation is most naturally performed in a branched diffusion model, where branch points have been explicitly defined and trained with. However, it is possible to perform a transmutation-like procedure in a label-guided (linear) model by first forward diffusing to some intermediate point (which we will call a "turn-back point"), and then reverse-diffusing from that point while supplying a desired target label. **a)** We show a random sample of 4s and 9s from MNIST, and perform transmutation using a branched diffusion model, with a well-defined branch point. The resulting digits show that transmutation in the branched model was both efficacious (i.e. the digit was successfully transformed from one class to the other) and analogous (i.e. non-class-specific features like slantedness were preserved). We then attempted transmutation in a linear model, using various turn-back points. Notably, in a linear model, the appropriate turn-back point is not known ahead of time (unlike in a branched model). As such, it is difficult to select the optimal turn-back point. Turn-back points which are too early cause transmutation to fail at efficacy: the target class is not generated at all. Turn-back points which are too late cause transmutation to fail at analogy: non-class-specific features like slantedness are no longer preserved. Additionally, some turn-back points (e.g. $t = 0.25$) cause some objects to be transmuted efficaciously, and others to fail to generate the target class entirely. Furthermore, even by setting the turn-back point to be the branch point in the branched model ($t = 0.35$), the transmuted results from the linear model are lower quality than in the branched model, likely because of crosstalk between the classes which is not controlled for at all in the training of the linear model.

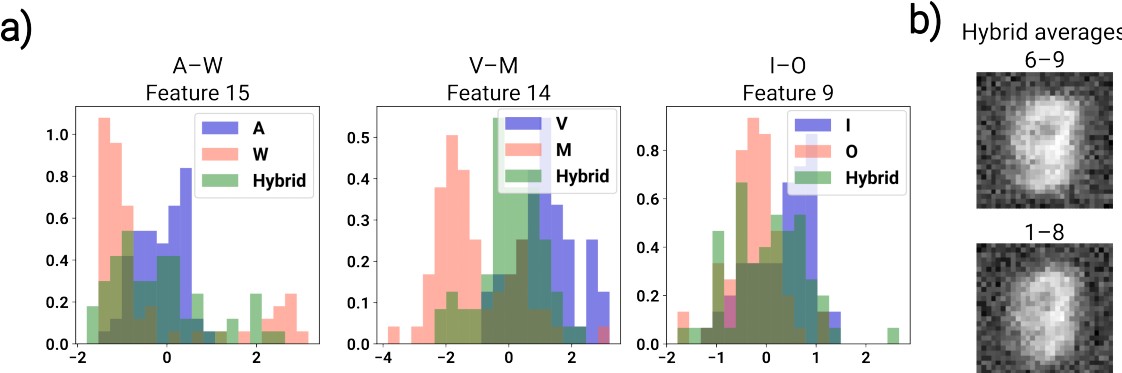

Figure S9: Interpreting hybrids of less-related classes. Branch points between less-related classes can still be interpreted, but between very unrelated classes, the interpretations are naturally less meaningful, as only very high-level (and relatively uninformative) features will be shared between these classes. **a)** We show the distribution of feature values for highly distinct features between less similar letters. The hybrids show a feature-value distribution which is intermediate and interpolated between the two classes, exhibiting properties of both. **b)** We show the hybrid resulting from interpreting the branch point between 6s and 9s or between 1s and 8s (dissimilar digits which were selected due to their late branch points) from our branched model trained on MNIST. The resulting hybrids show the common features between each pair of digit classes, which are generally fairly high-level, including: 1) the digits are in the center of the image; and 2) areas which are generally empty, coinciding with the "holes" of how most people draw their 6s or 9s, or their 8s.

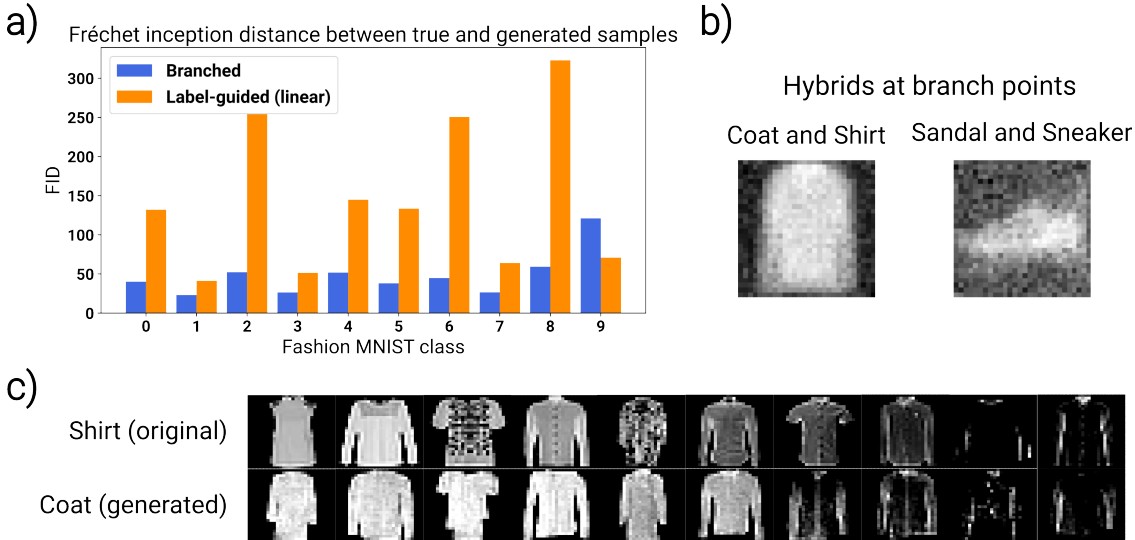

Figure S10: Branched models on Fashion MNIST. In addition to MNIST, we also show additional results on branched diffusion models trained on a more complex image dataset, but which does not suffer from the problem of centering. **a)** We found significantly better performance in all classes (except for one) compared to the linear model, and this difference was oftentimes stark. **b)** We show hybrids between certain classes, which show shared shapes between certain types of clothing. **c)** Transmuting between shirts and coats, certain analogous features are preserved, such as overall color (light vs dark) and relative sleeve length (shorter vs longer).

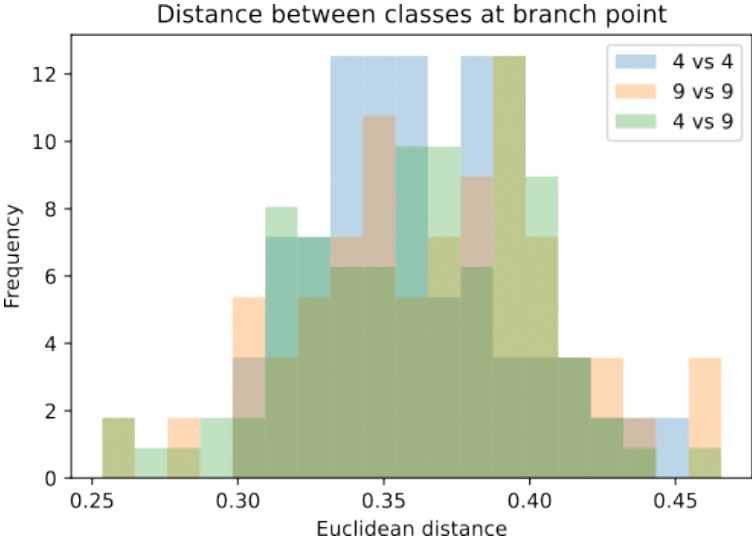

Figure S11: Distribution of distance between objects at branch point. We verify that distances between noisy objects of the *same* class are roughly the same compared to distances between noisy objects of *different* classes at the branch points. Here, we show the distance between noisy 4s, between noisy 9s, and between noisy 4s and 9s at the branch point. The distributions are almost identical, signifying that it takes no more "work" or "effort" for the diffusion model to generate various objects of the same class (e.g. 4s or 9s) compared to objects of different classes (e.g. 4s vs 9s). The observation that these distributions match is a direct consequence of how branch points are defined (Equation 8).

# D   Supplementary Methods

We trained all of our models and performed all analyses on a single Nvidia Quadro P6000.

## D.1   Training data

We downloaded the MNIST dataset and used all digits from `http://yann.lecun.com/exdb/mnist/` (LeCun et al.). We rescaled and recentered the values from [0, 256) to [-1, 1). This rescaling and symmetrization about 0 were to assist in the forward-diffusion process, which adds noise until the distribution approaches a standard (0-mean, identity-covariance) Gaussian.

We used the Fashion MNIST dataset as loaded from TorchVision.

We downloaded the tabular letter-recognition dataset from the UCI repository: `https://archive.ics.uci.edu/ml/datasets/Letter+Recognition` (Frey & Slate, 1991). We centered and scaled each of the 16 tabular features to zero mean and unit variance (pooled over the entire dataset, not for each individual letter class).

We downloaded the single-cell RNA-seq dataset from GEO (GSE149689) (Lee et al., 2020). We used Scanpy to pre-process the data, using a standard workflow which consisted of filtering out low-cell-count genes and low-gene-count cells, filtering out cells with too many mitochondrial genes, and retaining only the most variable genes and known marker genes (Wolf et al., 2018). We assigned cell-type labels using CellTypist (Conde et al., 2022). Of the annotated cell types, we retained 9 non-redundant cell types for training: CD16+ NK cells, classical monocytes, late erythrocytes, macrophages, megakaryocytes/platelets, memory B cells, NK cells, plasmablasts, and TEM/effector helper T cells. After pre-processing, the dataset consisted of 37102 cells (i.e. data points) and 280 genes (i.e. features). To train our diffusion models, we projected the gene expressions down to a latent space of 200 dimensions, using the linearly decoded variational autoencoder in scVI (Gayoso et al., 2022). The autoencoder was trained for 500 epochs, with a learning rate of 0.005.

We downloaded the ZINC250K dataset and converted the SMILES strings into molecular graphs using RDKit. We kekulized the graphs and featurized according to (Jo et al., 2022). We explored two methods of labeling the molecules for branched diffusion. First, we labeled molecules based on whether they were acyclic or had one cycle (molecules with multiple cycles were removed for simplicity). Secondly, we labeled molecules based on whether or not they possessed a halogen element (i.e. F, Cl, Br, I).

## D.2   Diffusion processes

For all of our continuous-time diffusion models, we employed the "variance-preserving stochastic differential equation" (VP-SDE) (Song et al., 2021). We used a variance schedule of $\beta(t) = 0.9t + 0.1$. We set our time horizon $T = 1$ (i.e. $t \in [0, 1)$). This amounts to adding Gaussian noise over continuous time. Our ZINC250K models were an exception, and we used the same diffusion processes (different for the node features and adjacency matrix) that were used in (Jo et al., 2022).

For our discrete-time diffusion model, we defined a discrete-time Gaussian noising process, following Ho et al. (2020). We defined $\beta_t = (1 \times 10^{-4}) + (1 \times 10^{-5})t$. We set our time horizon $T = 1000$ (i.e. $t \in [0, 1000]$).

## D.3   Defining branches

To discover branch points, we applied our branch-point discovery algorithm (Appendix A).

For our continuous-time branched model on MNIST (and Fashion MNIST), we used $\epsilon = 0.005$. For our discrete-time branched model on MNIST, we used $\epsilon = 0.001$. For our continuous-time branched model on tabular letters, we used $\epsilon = 0.01$. For our single-cell RNA-seq dataset, we used $\epsilon = 0.005$. These values were selected such that the branch points were not all too close to 0 or $T$.

The final branch definitions can be found in the Supplementary Figures and Tables.

For our ZINC250K dataset, we always used a branch point of 0.15.

## D.4  Model architectures

Our model architectures are designed after the architectures presented in Song et al. (2021) and Kotelnikov et al. (2022).

Our MNIST and Fashion MNIST models were trained on a UNet architecture consisting of 4 downsampling and 4 upsampling layers. In our branched models, the upsampling layers were shared between output tasks. We performed group normalization after every layer. The time embedding was computed as $[\sin(2\pi\frac{t}{T}), \cos(2\pi\frac{t}{T})]$. For each layer in the UNet, the time embedding was passed through a separate dense layer (unique for every UNet layer) and concatenated with the input to the UNet layer. For a label-guided model, we learned an embedding for each discrete label. As with the time embedding, the label embedding was passed through a separate dense layer (unique for every UNet layer) and concatenated to the input to each UNet layer.

Our letter models were trained on a dense architecture consisting of 5 dense layers. In our branched models, the first two layers were shared between output tasks. The time embedding was computed as $[\sin(2\pi\frac{t}{T}z), \cos(2\pi\frac{t}{T}z)]$, where $z$ is a set of Gaussian parameters that are not trainable. The time embeddings were passed through a dense layer, and the output was added to the input after the first dense layer. For a label-guided model, we again learned an embedding for each discrete label. The label embedding was passed through a dense layer, and the result was concatenated to the summation of the time embedding and the input after the first layer, before being passed to the remaining 4 layers.

Our single-cell RNA-seq models were trained on a dense residual architecture consisting of 5 dense layers. Each dense layer has 8192 hidden units. The input to each dense layer consisted of the sum of all previous layers' outputs. In our branched models, the first 3 layers were shared between output tasks. The time embedding was computed as $[\sin(2\pi\frac{t}{T}z), \cos(2\pi\frac{t}{T}z)]$, where $z$ is a set of Gaussian parameters that are not trainable. At each layer (other than the last), the time embedding was passed through a layer-specific dense mapping and added to the input to that layer. For a label-guided model, we again learned an embedding for each discrete label. The label embedding was passed through a dense layer and added to the very first layer's input.

Our ZINC250K branched models were trained on an architecture almost identical to that presented in Jo et al. (2022). In order to multi-task the model, we duplicate the last layers of the score networks for the node features or the adjacency matrix. For the node-feature score network, we multi-tasked only the final MLP layers. For the adjacency-matrix score network, we multi-tasked the final `AttentionLayer` and the final MLP layers. To incorporate $t$, we computed a time embedding as $[\sin(2\pi\frac{t}{T}z), \cos(2\pi\frac{t}{T}z)]$, where $z$ is a set of Gaussian parameters that are not trainable. This embedding was passed through a dense layer which projected this embedding down to a scalar, which was directly multiplied onto the final output of the score networks.

These neural-network architectures described above are the standard architectures used for linear (traditional) diffusion models for the associated data type (e.g. images, tabular data, or molecules). In order to turn these neural network architectures into multi-task architectures for branched diffusion models, we converted each model by multiplexing the last few layers (literally copying the last few layers $2|C| - 1$ times for $|C|$ classes), thereby turning the model into a multi-task model. The exact number of multiplexed layers multiplexed naturally depends on the data type and architecture, and is discussed below. This multiplexing was done such that the prediction path from the input to any single output head is architecturally identical to the standard single-task architecture.

For MNIST and Fashion MNIST, we multiplexed the last 4 layers of the U-Net (i.e. the upsampling layers). Note that because the U-Net concatenates downsampling (early) layers' outputs to the upsampler inputs, we effectively duplicate the downsamplers' outputs multiple times, as well, to feed into each output head. For letters, we multiplexed the last 2 layers. For RNA-seq, we multiplexed the last 3 layers. For molecules, we multiplexed the last 4 layers for the adjacency-matrix score, and the last 2 layers for the node-features score.

### D.5   Training schedules

For all of our models, we trained with a batch size of 128 examples, drawing uniformly from the entire dataset. This naturally ensures that branches which are longer (i.e. take up more diffusion time) or are responsible for more classes are upweighted appropriately.

For all of our models, we used a learning rate of 0.001, and trained our models until the loss had converged.

For our label-guided MNIST model, we trained for 30 epochs. For our label-guided letter model, we trained for 100 epochs. For our label-guided single-cell RNA-seq model, we trained for 100 epochs. In all cases, we noted that the loss had converged after training.

For our branched continuous-time MNIST model, we trained for 90 epochs. For our branched discrete-time MNIST model, we trained for 200 epochs. For our branched letter model (continuous-time), we trained for 100 epochs. For our branched single-cell RNA-seq model, we trained for 120 epochs. For our branched ZINC250K model labeled by cyclicity, we trained for 200 epochs; for our branched ZINC250K model labeled by halogenation, we trained for 50 epochs. Again, we noted that the loss had converged after training. Our branched model on Fashion MNIST used the same training procedure as with MNIST.

For our analysis on extending branched models and label-guided (linear) models to new classes, we also trained MNIST models on a subset of the dataset (i.e. only 0/4/9 or only 0/4/7/9), and single-cell RNA-seq models on a subset of the dataset (i.e. only CD16+ NK/classical monocytes or only CD16+ NK/classical monocytes/memory B cells). In these cases, we followed the same training parameters as above, except we trained for fewer epochs. In the class-extension analysis on MNIST, we started with branched or label-guided models trained on 0s, 4s, and 9s. These models we trained for 30 epochs each. In the class-extension analysis on single-cell RNA-seq, we started with a branched or label-guided model trained on CD16+ NK and classical monocytes. The branched model was trained for 120 epochs, and the label-guided model was trained for 100 epochs.

### D.6   Sampling procedure

When generating samples from a continuous-time diffusion model, we used the predictor-corrector algorithm defined in Song et al. (2021), using 1000 time steps from $T$ to 0. For our discrete-time diffusion model, we used the sampling algorithm defined in Ho et al. (2020). Note that we employed Algorithm 2 for branched models.

### D.7   Analyses

**Sample quality**

We compared the quality of samples generated from our branched diffusion models to those generated by our label-guided (linear) diffusion models using Fréchet Inception Distance (FID). For each class, we generated 1000 samples of each class from the branched model, 1000 samples of each class from the linear model, and randomly selected 1000 samples of each class from the true dataset. We computed FID over these samples, comparing each set of generated classes against the true samples. For the tabular letters dataset, there were not enough letters in the dataset to draw 1000 true samples of each letter, so we drew 700 of each letter from the true dataset. For the single-cell RNA-seq dataset, we generated 500 of each cell type from the diffusion models, and we sampled as many of each cell as possible from the true dataset (up to a maximum of 500).

**Class extension**

For our MNIST dataset, we started with a branched diffusion model trained on 0s, 4s, and 9s. To extend a new branch to reverse diffuse 7s, we simply created a new model with one extra output task and copied over the corresponding weights. For the new branch, we initialized the weights to the same as those on the corresponding 9 branch (with $s_i = 0$). We trained this new branch on only 7s, only for the time interval of that new branch. On the corresponding label-guided (linear) model, we fine-tuned on only 7s or on 0s, 4s, 7s, and 9s. In each experiment, we started with the linear model trained on 0s, 4s, and 9s.

For our single-cell RNA-seq dataset, we repeated the same procedure, but started with a branched diffusion model trained on CD16+ NK cells and classical monocytes, and introduced memory B cells as a new task. We trained the new branch on only memory B cells, only for the time interval of the new branch. On the corresponding label-guided (linear) model, we fine-tuned on only memory B cells or on all three cell types. To fine-tune on only memory B cells, we started with the linear model trained on CD16+ NK cells and classical monocytes only. To fine-tune on all three cell types, we again started with the linear model trained only on the original two cell types.

As above, we always fine-tuned until the loss converged. We note that this took much longer for the linear models compared to the branched model.

**Hybrid intermediates and transmutation**

For certain pairs of MNIST digits or letters, we found the earliest branch point for which they belong to the same branch, and generated hybrids by reverse diffusing to that branch point. To generate the average MNIST hybrids, we sampled 500 objects from the prior and reverse diffused to the branch point, and averaged the result.

Transmuted objects were computed by forward diffusing from one class to this branch point, and then reverse diffusing down the path to the other class from that intermediate.

To compute the preservation of functional groups in the transmutation of ZINC250k, we used the following list of functional groups:

```
https://github.com/Sulstice/global-chem/blob/development/
    global_chem/global_chem/miscellaneous/open_smiles.py
```

**Multi-class sampling efficiency**

We computed the amount of time taken to generate 64 examples of each class from our branched diffusion models, with and without taking advantage of the branch points. We took the average time over 10 trials each.

When leveraging the branching structure to generate samples, we ordered the branches by start time $s_i$ in *descending* order. For each branch $b_i$ in that order, we reverse diffused down the branch, starting with a cached intermediate at $t_i$ for the branch that ended at $t_i$. For the very first branch (the root), we started reverse diffusion by sampling $\pi(x)$. This guarantees that we will have a cached batch of samples at every branch point before we encounter a branch that starts at that branch point. Eventually, this algorithm generates a batch of samples for each class. For each branch, we performed reverse diffusion such that the total number of steps for any one class from $t = T$ to $t = 0$ was 1000.

To generate samples without leveraging the branching structure, we simply generated each class separately from the branched model, without caching any intermediates. Note that this takes the same amount of time as a purely linear model (of identical capacity and architecture) without any branching structure.

**Robustness of branch points**

To quantify the robustness of branched diffusion models to the underlying branch points, we computed the branch points for MNIST (continuous-time, all 10 digits) 10 times, each time following the procedure in Appendix A. Variation in the branch points resulted from variation in the randomly sampled objects, and in the forward-diffusion process. For each set of branch definitions, we trained a branched diffusion model using the procedure above. We then computed FID using the same procedure as above for other MNIST models, and compared the values to the FIDs of the corresponding label-guided MNIST model.

To quantify the similarity of the hierarchies, we computed the pairwise branch-score distance between all $\binom{10}{2}$ pairs of hierarchies discovered from our algorithm. We then generated 10 random hierarchies in a greedy fashion: start with all classes, and uniformly pick a random partition; uniformly pick a branch point $b_t$ between 0 and 1; recursively generate the two hierarchies below with a maximum time of $b_t$, until all class sets have been reduced to singletons. We used a Wilcoxon test to compare the distribution of branch-score

distances between the hierarchies discovered by our proposed algorithm, and the distances between the random hierarchies. Branch-score distance was computed using PHYLIP (Felsenstein).

