# OpenReview forum: "Hierarchically branched diffusion models leverage dataset structure for class-conditional generation"
_TMLR — Accepted by TMLR_

### Review · Reviewer_GyoJ · 2024-05-10

**Summary Of Contributions:**

The paper outlines an approach to conditional modeling of a dataset with $\vert C \vert$ labels in which the similarity between instances is used to inform a specialization into  $2\vert C \vert -1$ heads,
each head being responsible for a branch in a tree, with he root being the single shared "trunk" near the noise limit and lower levels of the tree being incrementally more specialized diffusion processes.
This tree is constructed via similarities of the $\ell_2$ distance of an auxiliary classifier by defining the branch point as the point where inter-class and intra-class similarity are matched in expectation and using a monte-carlo estimate to compute the branch point.

This architecture is claimed to enable class extension (add a new class into the diffusion process without requiring retraining on *all* data) while adding a new head (assuming the new class is sufficiently similar to existing classes), analogy generation (diffusion until the branch point, then reverse diffusion into a new class) and interpretable hybrids at the branch points, yielding insights into class feature structure.

Experiments are performed on MNIST, a tabular letter dataset, a single-cell RNA-seq gene expression dataset and  ZINC250K, with ablations being performed to the specific branch point timing

**Audience:**

Yes

**Broader Impact Concerns:**

I think adding BICs is not required

**Claims And Evidence:**

Yes

**Requested Changes:**

RFC


for the appendix

1. evaluate your branch point estimation by analysing whether a https://en.wikipedia.org/wiki/Welch%27s_t-test or another suitable statistical test would yield the same branch point method (and its error rate=
2. clarify the architecture used and the parameter count of vanilla vs your method. Also, report compute and timings of all methods in appendix

for the main body and/or appendix

1. I am concerned about the lack of improvement on the COVID task, can you run this on e.g. fashion MNIT, cifar10 and ciaf100 and resized imagenet-1k to ablate over increasing complexity and number of classes, and also investigate scalability? If not possible to perform, at least analyze the complexity of this method
2. critical: clarify whether the branch point computation is done with margin of probability, l2 or latent space l2
3. an ablation over class extension with a hold-out class of varying similarity ot existing classes would be an interesting nice to have (i.e., add a very dissimilar class vs. a very similar class to compare corners of the process).

**Strengths And Weaknesses:**

\+ conceptually nice motivation

\+ diverse-ish experiments

\+ ablations over branch points

\++ class extensions and analogies are new capabilities

0 limited discussion of computational intensity

0 foundation of branch point a bit ad-hoc

\- model details of implementation of heads not clear from text

\- evaluation on low dimensional data mainly, covid data only "real" task and limited performance there

\-- possible inconsistency: main body talks about euclidean distance(in latent space), appendix talks about latent space  of a classifier *AND* the margin of prediction.

---

> ### Author Response · Authors · 2024-06-03
>
> Thank you to the reviewer for the well-thought-out and constructive feedback! We will address each point below.
>
> **Clarification on branch-point computation**
>
> In our initial manuscript, we attempted to explain some of the intuition surrounding the branch-point discovery algorithm to readers by drawing an analogy to classifier margins. Based on this review and others, however, we learned that this was more confusing than needed, and so below we provide the more direct motivation and justification for our branch-point algorithm. The manuscript has also been updated accordingly.
>
> The goal of our branch-point discovery algorithm is to find the branch point between any pair of classes $c_{1}$ and $c_{2}$. The branch point is the earliest point in diffusion time where $c_{1}$ and $c_{2}$ are “indistinguishable” from each other. Formally, indistinguishability occurs when the conditional distributions of these two classes are close enough that they can be modeled by a single reverse-diffusion process. That is, we want the branch point $t_{b}$ to be such that $q_{t_{b}}(x\vert c_{1}) \approx q_{t_{b}}(x\vert c_{2})$.
>
> Of course, this is a difficult condition to formally define and satisfy, and there are several challenges:
>
> 1. We only have access to samples of objects from each class’ conditional distribution.
> 2. The data can be high dimensional, and particularly because we only have limited samples, the curse of dimensionality emerges quickly, even with as few as 10 dimensions.
> 3. These noisy conditional distributions $q_{t}(x\vert c)$ are not tractable to compute or even represent, particularly at earlier diffusion times (when the distribution $q_{t}(x)$ is so close to the unknown and complex data manifold $q_{0}(x)$).
> 4. The noisy conditional distributions for two different classes will generally not be identical (until the time horizon $T$), and they slowly approach indistinguishability asymptotically.
> 5. In order to find the optimal $t_{b}$, we must compare $q_{t}(x\vert c_{1})$ and $q_{t}(x\vert c_{2})$ at potentially all time points along the diffusion timeline, as well as across all pairs of classes in the dataset, so this comparison needs to be extremely computationally efficient.
>
> Based on these challenges and desiderata for comparing  $q_{t}(x\vert c_{1})$ and $q_{t}(x\vert c_{2})$, we opted to compute the distance between these two conditional distributions using *energy distance* [1]. Energy distance compares the expected distance between objects of two different distributions,  to the average expected distance within the same distribution. We chose to use energy distance for several reasons:

---

> > ### Author Response · Authors · 2024-06-03
> >
> > - Energy distance (by definition) compares the expected distance between two different distributions to the expected distance *within* the distributions. This allows us to attain a measure of “relative indistinguishability”. By picking a time point when the energy distance is low, we ensure that the model upstream of the branch point has no more difficulty representing and learning both classes together, as it would to learn only a single class (as in a standard, linear diffusion model). This addresses Challenge 4.
> > - Energy distance is much more robust in high dimensions. In high dimensions, particularly with sparser samples, metrics such as MMD and significance testing such as t-tests can be unreliable [2, 3]. This addresses Challenges 1 and 2.
> > - Energy distance does not require any distributional assumptions. Furthermore, we modify the energy distance formulation slightly, so that instead of taking the *difference* between expected inter-distribution and intra-distribution distances, we take the *ratio*, as this allows for the energy distance to be more comparable in magnitude across different classes and times, even as the distributions change drastically across diffusion time. It also allows us to select $\epsilon$ in a more principled and comparative manner across different datasets. In contrast, most frameworks for significance testing relies on strong distributional assumptions, particularly in high dimensions. This addresses Challenge 3.
> > - Energy distance is also extremely efficient to compute. It involves computing expected distance between objects of the two distributions, as well as between objects of the same distribution. In our algorithm, we perform Monte Carlo sampling over possible random couplings between objects to compute these expected distances. This make energy distance far more suitable for our needs compared to other metrics such as Wasserstein distance, which requires minimizing over all possible couplings. This addresses Challenge 5.
> > - Energy distance is calculated simply using Euclidean distance, and there is intuitive reason for why Euclidean distance in diffusion space is a meaningful measurement of distance between distributions. The Euclidean distance can measure how much “work” a diffusion model does to push objects from one point to another. In particular, we can view the reverse-diffusion process as taking a sampled object from $\pi(x)$ and pushing the feature values to a final generated sample. For a single branch representing multiple classes over some diffusion time interval, distances between objects reflect the expected distance the branch will be pushing objects, and we would like this distance to be no more for objects of different classes versus objects of the same class in that branch.
> >
> > Note that the L2 distance is calculated in whichever space the diffusion model performs diffusion. For example, if the diffusion is on direct images, we take L2 distance between noisy images (averaged over the pixels). If the diffusion is on latent-space representations, we take L2 distance between noisy latent-space representations. This ensures that the branch points are precisely tailored to the diffusion process and dataset that are being used. That is, *branch-point computation is done using L2 distance on whatever the input to the diffusion model is.*
> >
> > Of course, there is some variability in the possible branch points that arise from this algorithm, and therefore we also analyzed robustness of our models to branch-point variations (Supplementary Figures S5—S6).
> >
> > Regarding the use of statistical tests for branch-point discovery, we opted for a metric such as energy distance, because statistical tests (e.g. Welch’s t-test) can be far less powerful or meaningful in high dimensions, particularly when there are fewer samples (Challenges 1 and 2) [2, 3]. These tests are also oftentimes extremely computationally demanding, and would not be suitable for our purposes (Challenge 5) [3]. Furthermore, other statistical tests which are more suited for higher dimensions (e.g. MANOVA, quadratic or linear discriminant analysis, etc.) typically require distributional assumptions which are very much not upheld, particularly at early diffusion times (Challenge 3).
> >
> > The manuscript has been updated with these details.

---

> > > ### Author Response · Authors · 2024-06-03
> > >
> > > **Details on implementation of model output heads**
> > >
> > > In our work, we applied branched diffusion models to a variety of datasets spanning various data modalities and complexities. For each dataset (MNIST, letters, RNA-seq, and molecules), we used a different neural-network architecture to learn diffusion, as each dataset had its own requirements for the optimal neural-network architecture (e.g. for MNIST, we used a U-Net; for molecules, we used a graph convolutional network).
> > >
> > > For each of these models, we began with the standard architecture used for the linear (traditional) diffusion model, and converted it to a branched model by multiplexing the last few layers (literally copying the last few layers $2\vert C\vert - 1$ times for $\vert C\vert$ classes), thereby turning the model into a multi-task model. The exact number of multiplexed layers multiplexed naturally depends on the data type and architecture, and is discussed below. This multiplexing was done such that the prediction path from the input to any single output head is architecturally identical to the standard single-task architecture.
> > >
> > > The single-task architecture is described in the supplementary methods.
> > >
> > > For MNIST, we multiplexed the last 4 layers of the U-Net (i.e. the upsampling layers). Note that because the U-Net concatenates downsampling (early) layers’ outputs to the upsampler inputs, we effectively duplicate the downsamplers’ outputs multiple times, as well, to feed into each output head. For letters, we multiplexed the last 2 layers. For RNA-seq, we multiplexed the last 3 layers. For molecules, we multiplexed the last 4 layers for the adjacency-matrix score, and the last 2 layers for the node-features score.
> > >
> > > We updated the manuscript to better clarify this point.
> > >
> > > **Dimensionality of data**
> > >
> > > To clarify the dimensionality and complexity of our datasets, we have datasets ranging from few classes to 26 classes (letters). The larger dimensionalities are the RNA-seq dataset (200 latent dimensions), MNIST (784 dimensions), and molecules (graphs of up to 44 heavy atoms). Our dataset sizes also range from a few thousand examples (letters), to tens of thousands (RNA-seq) or hundreds of thousands (molecules).
> > >
> > > **Complexity of model architectures and training times versus dataset complexity**
> > >
> > > We provide the following quantitative results for model complexity and training time:
> > >
> > > | Dataset | Number of classes | Total branch length | Relative model capacity | Relative number of epochs |
> > > | --- | --- | --- | --- | --- |
> > > | MNIST | 10 | 11.49 | 8.48 | 3 |
> > > | Fashion MNIST | 10 | 4.81 | 8.48 | 2 |
> > > | Letters | 26 | 10.05 | 11.06 | 1 |
> > > | RNA-seq | 9 | 4.29 | 6.38 | 1.2 |
> > > | ZINC250K | 2 | 1.15 | 1.89 | 1 |
> > >
> > > We report the model capacity as the ratio of the number of parameters relative to a non-multi-tasked (traditional, linear) diffusion model. In order to illustrate the computational intensity of branched diffusion models, we also report the number of epochs trained, relative to the number of epochs in the analogous linear model. Note that both space and time complexity grow sublinearly relative to the dataset size in terms of number of classes.
> > >
> > > We also report the total branch length, which is the total amount of diffusion time represented by the branched diffusion model (i.e. the sum of the total branch lengths in diffusion time). In a traditional linear model, the total branch length is 1. Note that in both a linear and a branched diffusion model, the amount of diffusion time taken to generate any single example is 1.
> > >
> > > In general, the increased model complexity depends on the number of classes, as well as the overall similarity of the classes to each other (this is very dataset dependent). Note that the total branch length is a measure of this overall similarity. If all classes are very similar, the total branch length will be close to 1; if many classes are very distinct from each other, the branch points will be later in diffusion time and lead to a higher total branch length.
> > >
> > > We found that in general, the time complexity did not change significantly. For image datasets, however (e.g. MNIST), we did find we needed to train for slightly longer, and this is likely due to the inherent centering challenge with images (as described in the Discussion section of our manuscript).
> > >
> > > Strategies to reduce this overhead are possible, for example, by merging very similar tasks/heads or increasing parameter sharing. We plan to investigate possible extensions in future work.
> > >
> > > This table has been added to the manuscript as Supplementary Table S10.

---

> > > > ### Author Response · Authors · 2024-06-03
> > > >
> > > > **Tests on Fashion MNIST**
> > > >
> > > > In addition to the complexity analysis above, we also present novel results on branched and linear models trained on Fashion MNIST. Because Fashion MNIST consists of largely centered images, the problem of centering is not as severe, and there was no need to perform diffusion in latent space.
> > > >
> > > > We ran the branch-point discovery algorithm on this dataset using the same sample sizes and $\epsilon$ as with MNIST, and trained a branched diffusion model on those branch points. In terms of performance, we found significantly better performance in all classes (except for one) compared to the linear model, and this difference was oftentimes stark. Note that the training details (including model architectures and training epochs) matched those we used with MNIST (i.e. we did not perform any specific fine-tuning or optimization for our models here), although we found that both models required fewer epochs to converge compared to MNIST.
> > > >
> > > > Of course, we can also apply techniques unique to branched diffusion models, such as transmutation. For example, transmuting between shirts and coats, certain analogous features are preserved, such as overall color (light vs dark) and relative sleeve length (shorter vs longer) (Supplementary Figure S10c).
> > > >
> > > > We can also visualize hybrids between objects, but they are not as interesting as with other considered datasets (Supplementary Figure S10b).
> > > >
> > > > These results have been added to the manuscript as Supplementary Figure S10.
> > > >
> > > > **Class extension on varying similarity to existing classes**
> > > >
> > > > In our supplement, we performed an ablation over class extension where the hold-out class was of varying similarity to existing classes.
> > > >
> > > > In Supplementary Figure S7, we performed multi-step continual-learning where we extended our MNIST models to three never-before-seen classes one by one: 7s, 1s, then 2s (a mixture of classes which are similar to or distinct from existing classes in the model). These classes range in difficulty in the sense that these novel digits range in similarity to existing digits in the model (i.e. 7s are most similar to the existing classes, 1s are moderately similar, and 2s are very dissimilar). As we added multiple novel classes ranging in similarity/difficulty to existing data, we found that the benefits offered by branched diffusion models for continual learning are consistent.
> > > >
> > > > **Predictive performance on RNA-seq**
> > > >
> > > > For the RNA-seq dataset, below we reproduce the FID values for each class. Due to the high variation in FID values between the classes (likely because some cell types are inherently harder to generate than others), differences in these values were not obviously rendered in Supplementary Figure S3. However, we did see a significant improvement in most classes using the branched model (i.e. *performance did improve with the branched model*).
> > > >
> > > > |  | Branched | Label-guided |
> > > > | --- | --- | --- |
> > > > | CD16+ NK | 377.254 | 376.300 |
> > > > | Cl. Mono. | 163.470 | 166.414 |
> > > > | Late Eryth. | 150.605 | 153.134 |
> > > > | Macroph. | 151.118 | 156.442 |
> > > > | Megakar. | 64.968 | 65.506 |
> > > > | Mem. B | 1604.477 | 1610.171 |
> > > > | NK | 84.447 | 85.517 |
> > > > | Plasmabl. | 317.073 | 317.584 |
> > > > | Tem/Eff. H. T | 174.465 | 175.692 |
> > > >
> > > > **References**
> > > >
> > > > [1] Székely and Rizzo. Energy statistics: A class of statistics based on distances. (2013) https://doi.org/10.1016/j.jspi.2013.03.018.
> > > >
> > > > [2] Reddi, Ramdas, et. al. On the High Dimensional Power of a Linear-Time Two Sample Test under Mean-shift Alternatives. (2015) https://www.cs.cmu.edu/~aarti/pubs/AISTATS15_SReddiARamdas.pdf
> > > >
> > > > [3] Bischoff, et. al. A Practical Guide to Statistical Distances for Evaluating Generative Models in Science. (2024) https://arxiv.org/pdf/2403.12636v1

---

### Review · Reviewer_nEk7 · 2024-05-14

**Summary Of Contributions:**

The paper introduces a novel diffusional model that incorporates branching to facilitate class-conditional sampling. The paper provides brief algorithms for sampling and training (assuming that the branch structure is known). A heuristic for identifying the branch structure is provided in an appendix. The paper presents experimental results for multiple datasets of different modalities. First, there is a demonstration that the proposed branched model can achieve generative performance that is equivalent to (or perhaps even slightly better than) a traditional model. Second, the paper illustrates how the model can incorporate new classes without requiring complete re-training. Third, the paper provides examples of conditional generation via transmutation. Finally, the paper discusses interpretability of the branchpoints and the potential for efficient multi-class sampling.

**Audience:**

Yes

**Broader Impact Concerns:**

No concerns

**Claims And Evidence:**

Yes

**Requested Changes:**

(1)	[Critical] The main body of the paper should include some discussion of the choice of branch points in the methodology section. It is fine to leave the details to an appendix, but there should be acknowledgement in the methodology that this is a critical component of using such a model (and underpins all of the presented results.

(2)	[Critical] The decision to base the branching point on indistinguishability according to a linear classifier appears odd. This would seem to be a low bar in the sense that the distributions could still be structurally very different (e.g, circle enclosed by a ring) and yet the linear classifier would struggle. Please add some text that provides a better motivation for this design decision. In particular, why is it not possible to use more standard approaches for testing whether two distributions are equivalent? (e.g., something like an MMD test, such as the one in Schrab et al., “MMD Aggregated Two-Sample Test”, JMLR 2023). Is there a reason that the diffusion process means that the classifier (Euclidean distance) is a sensible strategy?

(3)	[Recommended, not mandatory] The paper includes experimental results in Section 3.1, a subsection of Section 3, which describes the model and algorithms. I think the paper would be easier to digest if the reader can separate the methodology from the experimental work, and I would recommend moving Section 3.1 to Section 4. Sections 3.1, 4 and 5 could be subsections of an “Applications and Experiments” section. Section 7 is very brief and contains very little information. If it is not going to present any results, or even a summary of results, then it should be included as a paragraph in the Discussion section.

**Strengths And Weaknesses:**

Strengths

(1)	The model proposed in the paper is novel and is an elegant and intuitively-appealing strategy for performing class-conditional sampling using a diffusion model.

(2)	The experiments use datasets from multiple modalities and illustrate the potential of the proposed approach. In addition to achieving good generative performance, the method has appealing capabilities to incorporate new classes or perform transmutation.

(3)	The paper is well-written and describes the technique and experiments clearly (although there are some structural issues and some important information is excluded from the main body).

Weaknesses

(1)	The model relies on the selection of good branchpoints (although the paper does include some results exploring robustness). The paper presents the model as though the branchpoints are provided as input, but based on all of the presented experiments, it appears that the branchpoints are actually parameters of the model. All of the results rely on a procedure to choose these branchpoints, but that procedure is not discussed in the main text of the paper.

(2)	The branch selection procedure incorporates design choices that are not particularly well-motivated (e.g., linear classifier, Euclidean distance).

(3)	Some of the structural choices of the paper could be improved with a view to facilitating comprehension for a reader.

---

> ### Author Response · Authors · 2024-06-03
>
> Thank you to the reviewer for the well-thought-out and constructive feedback! We will address each point below.
>
> **Clarification on branch-point discovery**
>
> In our initial manuscript, we attempted to explain some of the intuition surrounding the branch-point discovery algorithm to readers by drawing an analogy to classifier margins. Based on this review and others, however, we learned that this was more confusing than needed, and so below we provide the more direct motivation and justification for our branch-point algorithm. The manuscript has also been updated accordingly.
>
> The goal of our branch-point discovery algorithm is to find the branch point between any pair of classes $c_{1}$ and $c_{2}$. The branch point is the earliest point in diffusion time where $c_{1}$ and $c_{2}$ are “indistinguishable” from each other. Formally, indistinguishability occurs when the conditional distributions of these two classes are close enough that they can be modeled by a single reverse-diffusion process. That is, we want the branch point $t_{b}$ to be such that $q_{t_{b}}(x\vert c_{1}) \approx q_{t_{b}}(x\vert c_{2})$.
>
> Of course, this is a difficult condition to formally define and satisfy, and there are several challenges:
>
> 1. We only have access to samples of objects from each class’ conditional distribution.
> 2. The data can be high dimensional, and particularly because we only have limited samples, the curse of dimensionality emerges quickly, even with as few as 10 dimensions.
> 3. These noisy conditional distributions $q_{t}(x\vert c)$ are not tractable to compute or even represent, particularly at earlier diffusion times (when the distribution $q_{t}(x)$ is so close to the unknown and complex data manifold $q_{0}(x)$).
> 4. The noisy conditional distributions for two different classes will generally not be identical (until the time horizon $T$), and they slowly approach indistinguishability asymptotically.
> 5. In order to find the optimal $t_{b}$, we must compare $q_{t}(x\vert c_{1})$ and $q_{t}(x\vert c_{2})$ at potentially all time points along the diffusion timeline, as well as across all pairs of classes in the dataset, so this comparison needs to be extremely computationally efficient.
>
> Based on these challenges and desiderata for comparing  $q_{t}(x\vert c_{1})$ and $q_{t}(x\vert c_{2})$, we opted to compute the distance between these two conditional distributions using *energy distance* [1]. Energy distance compares the expected distance between objects of two different distributions,  to the average expected distance within the same distribution. We chose to use energy distance for several reasons:

---

> > ### Author Response · Authors · 2024-06-03
> >
> > - Energy distance (by definition) compares the expected distance between two different distributions to the expected distance *within* the distributions. This allows us to attain a measure of “relative indistinguishability”. By picking a time point when the energy distance is low, we ensure that the model upstream of the branch point has no more difficulty representing and learning both classes together, as it would to learn only a single class (as in a standard, linear diffusion model). This addresses Challenge 4.
> > - Energy distance is much more robust in high dimensions. In high dimensions, particularly with sparser samples, metrics such as MMD and significance testing such as t-tests can be unreliable [2, 3]. This addresses Challenges 1 and 2.
> > - Energy distance does not require any distributional assumptions. Furthermore, we modify the energy distance formulation slightly, so that instead of taking the *difference* between expected inter-distribution and intra-distribution distances, we take the *ratio*, as this allows for the energy distance to be more comparable in magnitude across different classes and times, even as the distributions change drastically across diffusion time. It also allows us to select $\epsilon$ in a more principled and comparative manner across different datasets. In contrast, most frameworks for significance testing relies on strong distributional assumptions, particularly in high dimensions. This addresses Challenge 3.
> > - Energy distance is also extremely efficient to compute. It involves computing expected distance between objects of the two distributions, as well as between objects of the same distribution. In our algorithm, we perform Monte Carlo sampling over possible random couplings between objects to compute these expected distances. This make energy distance far more suitable for our needs compared to other metrics such as Wasserstein distance, which requires minimizing over all possible couplings. This addresses Challenge 5.
> > - Energy distance is calculated simply using Euclidean distance, and there is intuitive reason for why Euclidean distance in diffusion space is a meaningful measurement of distance between distributions. The Euclidean distance can measure how much “work” a diffusion model does to push objects from one point to another. In particular, we can view the reverse-diffusion process as taking a sampled object from $\pi(x)$ and pushing the feature values to a final generated sample. For a single branch representing multiple classes over some diffusion time interval, distances between objects reflect the expected distance the branch will be pushing objects, and we would like this distance to be no more for objects of different classes versus objects of the same class in that branch.
> >
> > Note that using energy distance as described here, it behaves as desired for the “circle enclosed by a ring” example brought up in the review. We agree that the linear-classifier analogy is in inadequate way of explaining the intuition, as the analogy fails for an example like this. We hope that our motivations for using energy distance are a better way of presenting the intuition behind our use of Euclidean distance.
> >
> > Regarding MMD, we opted to use energy distance instead of MMD because MMD is much more unreliable in high dimensions with potentially fewer samples (Challenges 1 and 2) [2], and it would be very difficult to choose a justifiable kernel given the lack of distributional assumptions we can make (Challenge 3) [3]. Furthermore, energy distance is far quicker to compute compared to MMD (Challenge 5).
> >
> > The manuscript has been updated with these details.
> >
> > **Improved structuring of writing**
> >
> > We have updated some of the writing in the main text and appendices to reflect all suggestions made.
> >
> > **References**
> >
> > [1] Székely and Rizzo. Energy statistics: A class of statistics based on distances. (2013) https://doi.org/10.1016/j.jspi.2013.03.018.
> >
> > [2] Reddi, Ramdas, et. al. On the High Dimensional Power of a Linear-Time Two Sample Test under Mean-shift Alternatives. (2015) https://www.cs.cmu.edu/~aarti/pubs/AISTATS15_SReddiARamdas.pdf
> >
> > [3] Bischoff, et. al. A Practical Guide to Statistical Distances for Evaluating Generative Models in Science. (2024) https://arxiv.org/pdf/2403.12636v1

---

### Review · Reviewer_wkP3 · 2024-05-28

**Summary Of Contributions:**

This paper introduces the idea of class-conditional hierarchical branching in diffusion models. Namely, there is a point in the diffusion generation process when the class of the example is decided, which we call the branching point. The authors try to find the average such point for each branch by considering when the objects of different classes are as different as objects of the same class. The authors then train such hierarchical branching diffusion models and show their application in different problems, in particular, continual learning, analogy-based conditional generation, and interpretability.

**Audience:**

Yes

**Claims And Evidence:**

No

**Requested Changes:**

Please address the issues with the evaluation and identification of the branching point described in the weaknesses section. I would say that having a better baseline in the CL setting is of critical importance and the discussion of how well we can identify the branching point is very much needed. The other points are nice to have but not that critical, in my opinion.

**Strengths And Weaknesses:**

Strengths:

- The hierarchical branching of diffusion models is an interesting idea that the authors test in diverse scenarios: continual learning, interpretability, and interpolation.
- Paper is clearly written and easy to read.
- The appendix is full of additional experiments and details.
- The supplementary material contains the code and several notebooks for reproducing the results.

Weaknesses:

- Evaluation is insufficient.
    - The datasets used in this paper are fairly simple. Diffusion models are primarily used with images, and the image experiments here are run only on MNIST, which is a trivial dataset. It is not apparent if this approach scales up to Cifar or CelebA. The ZINC250K and RNA-seq datasets are nice additions, but I still think an extra image dataset would be useful.
    - The baselines are not present or too weak. I don’t think we require the method to outperform all prior work but having some solid results for reference would be useful.
        - In continual learning experiments, the architecture for branched diffusion consists of a shared encoder and a single head per branch. Only the heads are trained on new tasks, the encoder is frozen. On the other hand, it seems like label-guided diffusion trains the whole model without any freezing. In the label-guided setting, one still could have a bunch of shared parameters followed by a single head per label — I would expect this to work much better.
        - In interpolation experiments, it would be nice to have some baselines. Interpolation with diffusion models is not the most trivial thing, but there have been several approaches for this purpose. [1, 2]
        - In the interpretability experiments, there is no reference method. I don't have any great suggestions here, but one could, for example, use the label-conditioned diffusion with the label set at `[0.5, 0.5]` between relevant classes.
- The technique for branching point identification is not under empirical scrutiny here. The authors describe the theoretical underlying of the approach in Appendix A, but there aren’t any experiments to check if the identified branching points $t_b$ work for all examples for a given task. I would imagine that for certain examples (noise samples), the class becomes visible much more quickly than for others, and even more so in complex images.
    - For example, if we were to apply the branching to Cifar-10 and find the branching point between classes “bird” and “airplane” then examples with blue background (might be either a sky or a bird against a sky) would branch much later than examples with a green background (probably a bird in a forest).


[1] Wang, Clinton, and Polina Golland. "Interpolating between images with diffusion models." (2023). \
[2] Jain, Siddhant, et al. "Video Interpolation with Diffusion Models." arXiv preprint arXiv:2404.01203 (2024).

---

> ### Author Response · Authors · 2024-06-05
>
> Thank you to the reviewer for the well-thought-out and constructive feedback! We will address each point below.
>
> **Baselines for continual learning**
>
> For traditional (label-guided) diffusion models, we showed that they are not amenable to continual learning, as it is not easy to introduce new classes/data and have them retain their ability to generate existing classes (Figure 2).
>
> In the current literature on diffusion models, class-conditional diffusion models are universally single-task neural networks, where the class label is fed in as an auxiliary input, very early in the architecture [1, 2, 3]. This is partially why extending to novel classes is particularly difficult for these models.
>
> On the other hand, one certainly could imagine another type of linear (non-branched) diffusion model where the neural network is multi-tasked, and each output task performs reverse diffusion for a single class, as suggested above in the review. We briefly proposed a similar setup in our manuscript, in the middle of Figure 1c. In this figure, we noted that one could train a class-conditional diffusion model using several neural networks, one for each class. The suggested setup above is simply the multi-task version of this structure, where the early layers are now shared between each neural network.
>
> As we noted in our manuscript, this method can be viewed as a middle-ground between traditional linear diffusion models and branched diffusion models. In particular, branched diffusion models also encode class identity using different output heads, but diffusion time is maximally shared, instead of having each output head predict the entire diffusion timeline for its class. We opted to forgo comparisons to this middle-ground method, as it is a completely novel method which does not exist in the literature (as far as we know). Additionally, although it would be naturally very amenable to continual learning, it would not have any of the other benefits of branched diffusion models, such as transmutation, interpretability, or efficient generation.

---

> > ### Author Response · Authors · 2024-06-05
> >
> > **Clarification on branch points for classes**
> >
> > Since branched diffusion models are a method for class-conditional generation, the branch points are defined as properties of entire classes of objects, rather than individual objects. Because class identity is encoded as a collection of output tasks of the neural network (which cover the entire reverse-diffusion timeline from $T$ to $0$), branch points are most naturally defined at the level of classes. As such, the branch points are defined based on the distance between *distributions* of noisy objects of each class.
> >
> > Importantly, the branch points do not force all individual objects to diffuse the same exact “distance” in feature space. Instead, the branch points define the amount of diffusion time for which labels are provided, so that the model can differentiate between certain classes (or groups of classes).
> >
> > For example, consider the two classes of birds and airplanes with a branch point at time $t_{b}$ between them. In this two-class example, there are three branches: a shared branch representing diffusion time $[t_{b}, T]$ for both classes, a bird-specific branch covering diffusion time $[0, t_{b}]$, and a plane-specific branch covering diffusion time $[0, t_{b}]$. The branch point at $t_{b}$ is not a direct measurement of how much “diffusion” distance every single bird or plane must undergo. Instead, $t_{b}$ is the amount of diffusion time required for the model to generate a bird or plane once the final label is known. That is, it takes $T - t_{b}$ diffusion time to start from random noise and construct the higher-order shared features between birds and planes (this is on average; of course, these higher-order shared features also follow a distribution over objects and diffusion noise). It then requires $t_{b}$ diffusion time to generate bird-specific or plane-specific features that are unique to those classes, as well as specific instances/images. In particular, it is only during these class-specific branches that the model has knowledge of the final label (i.e. that it is generating a bird or a plane). For example, to generate an image of a bird, a branched diffusion model has $T - t_{b}$ diffusion time to construct as many features as it can without knowledge that the final label is “bird”. Then it has $t_{b}$ diffusion time to construct features that generate a specific bird image. No matter what the background color of the final image is, the model has $t_{b}$ diffusion time to generate the final bird from the noisy image it starts with at time $t_{b}$.
> >
> > In many ways, we can also think of each branch in a branched diffusion model as a separate *unconditional* traditional (linear) diffusion model. In the bird-and-airplane example above, the bird-specific branch is a linear diffusion model, where we first sample objects from a prior distribution $q_{t_{b}}(x)$, and the model is trained to reverse-diffuse from this prior distribution to the distribution of birds. Of course, this prior distribution $q_{t_{b}}(x)$ is very intractable, as it has many bird-like features (and plane-like features, as well). In order to sample from this prior distribution at $t_{b}$, we train an additional linear diffusion model to reverse-diffuse from a tractable prior $\pi(x)$ (e.g. a Gaussian) to $q_{t_{b}}(x)$. $t_{b}$ is selected such that this linear diffusion model from $\pi(x)$ to $q_{t_{b}}(x)$ can be shared between both birds and planes. As such, a branched diffusion model can be thought of as a “hierarchy” of linear diffusion models.
> >
> > Notably, we could train a branched diffusion model where all of the branch points are at $T$. That is, every class gets the entire diffusion timeline from $T$ to $0$ where the label of the class is known. This special case of branched diffusion models is precisely the multi-tasked label-guided model which was proposed in this review (and discussed in the previous section of this response). As discussed above, a branch-point structure like this would likely be sufficient for predictive performance and continual learning, but would lose out on the other benefits of branched diffusion models if we had selected better branch points which maximally share diffusion time.
> >
> > It may also be helpful to point out that the performance of branched diffusion models is largely robust to variations in the branch points (Supplementary Figures S5—S6).
> >
> > We can also verify that distances between noisy objects of the *same* class are roughly the same compared to distances between noisy objects of *different* classes at the branch points. This means that it takes no more “work” or “effort” for the diffusion model to generate various objects of the same class (e.g. birds of blue vs green backgrounds) compared to objects of different classes (e.g. birds vs planes). This is not surprising, because this is precisely how branch points are defined.
> >
> > We have added this figure to our manuscript as Supplementary Figure S11.

---

> > > ### Author Response · Authors · 2024-06-05
> > >
> > > **Baselines for interpretability and transmutation**
> > >
> > > Even if not listed as critical in the review, we also investigated potential baselines for interpretability and transmutation to expand the discussion in our manuscript. As noted in the review, there is no “obvious” way to extend standard diffusion models with these capabilities.
> > >
> > > For the suggested baseline on the interpretability of hybrids, we attempted the suggestion of taking an average of label embeddings. However, the resulting generated objects were not accurate to the intended embedding. For example, on our MNIST dataset, we constructed a label embedding which is the average of the embeddings for 4s and 9s. Instead of getting “hybrid” objects, the generated images are either 4s or 9 (and not particularly high quality). We hypothesize that this is due to the conditioning embedding being out of distribution. We will add this experiment to the Appendix of the final version of the manuscript.
> > >
> > > For transmutation, it remains difficult to compare and evaluate. We thank the reviewer for pointing out these recent works on interpolation using diffusion (we actually did not see these works, as they were posted after we wrote our manuscript!). Notably, however, interpolation is a slightly different problem than transmutation. In our work, transmutation is most similar to neural style transfer, where we condition generation on a specific object, as well as a general class label (e.g. a starting molecule and the class label of “halogenated”). Transmutation and style transfer remain quite difficult to quantify and compare, and in our manuscript, we relied heavily on our scientific-domain knowledge to quantitatively verify the success of transmutation: we computed the extent to which the transmuted object loses class-specific features of the source class, gains class-specific features of the target class, and retains non-class-defining features (Figure 3). Note that the original neural-style-transfer works did not perform any quantitative analysis due to these challenges [4].
> > >
> > > **Evaluation on images**
> > >
> > > As noted in our Discussion, branched diffusion models do struggle more with image datasets due to the problem of centering. Notably, however, our main focus is on scientific applications and non-image data (e.g. RNA-seq and molecules), which we showed several examples of.
> > >
> > > In order to provide extra image results, however, we performed additional experiments on Fashion MNIST, which also avoids the problem of image centering, while providing more complex images and class structures. Without any fine-tuning, we found that branched diffusion models continued to offer the same benefits we saw in our other datasets, relative to traditional linear models, including performance, interpretability, and transmutation. We included these results in the manuscript as Supplementary Figure S10.
> > >
> > > **References**
> > >
> > > [1] Ho, et. al. “Classifier-free diffusion guidance” (2021).
> > >
> > > [2] Rombach, et. al. “High-resolution image synthesis with latent diffusion models” (2022).
> > >
> > > [3] Jo, et. al. “Score-based generative modeling of graphs via the system of stochastic differential equations” (2022).
> > >
> > > [4] Gatys, et. al. “A Neural Algorithm of Artistic Style” (2015).

---

### Decision · Action_Editor_SW9R · 2024-08-08

**Recommendation:** Accept as is

**Comment:**

Two reviewers recommended acceptance, one reviewer weakly opposed acceptance. The submission satisfies the TMLR criteria for acceptance.

**Audience:**

Diffusion models and continual learning.

**Claims And Evidence:**

Paper introduces class-conditional hierarchical branching in diffusion models. Evaluations include different data modalities (images, tabular data, and graphs) and scientific datasets.